# *Bordetella* adenylate cyclase toxin is a unique ligand of the integrin complement receptor 3

Radim Osicka[1]*, Adriana Osickova[1,2], Shakir Hasan[1], Ladislav Bumba[1], Jiri Cerny[3], Peter Sebo[1]

[1]Institute of Microbiology of the Czech Academy of Sciences, Prague, Czech Republic; [2]Department of Biochemistry, Faculty of Science, Charles University in Prague, Prague, Czech Republic; [3]Institute of Biotechnology of the Czech Academy of Sciences, Prague, Czech Republic

**Abstract** Integrins are heterodimeric cell surface adhesion and signaling receptors that are essential for metazoan existence. Some integrins contain an I-domain that is a major ligand binding site. The ligands preferentially engage the active forms of the integrins and trigger signaling cascades that alter numerous cell functions. Here we found that the adenylate cyclase toxin (CyaA), a key virulence factor of the whooping cough agent *Bordetella pertussis,* preferentially binds an inactive form of the integrin complement receptor 3 (CR3), using a site outside of its I-domain. CyaA binding did not trigger downstream signaling of CR3 in human monocytes and CyaA-catalyzed elevation of cAMP effectively blocked CR3 signaling initiated by a natural ligand. This unprecedented type of integrin-ligand interaction distinguishes CyaA from all other known ligands of the I-domain-containing integrins and provides a mechanistic insight into the previously observed central role of CyaA in the pathogenesis of *B. pertussis.*

*For correspondence: osicka@ biomed.cas.cz

**Competing interests:** The authors declare that no competing interests exist.

## Introduction

Integrins are dimeric transmembrane proteins complexes composed of an alpha and a beta subunit. There are 18 different alpha subunits and 8 beta subunits that combine in a limited number of combinations of which 24 are currently known in mammals (*Tan, 2012*). Integrins are essential for regulation of numerous cellular functions including cell signaling and adhesion. Nine of the eighteen integrin alpha subunits harbor a conserved I (inserted)-domain that is crucial for binding of endogenous ligands (*Johnson and Chouhan, 2014*). Four of these integrin alpha subunits ($\alpha_D$, $\alpha_L$, $\alpha_M$ and $\alpha_X$) form heterodimers exclusively with the $\beta_2$ subunit, thus forming the $\alpha_D\beta_2$ (CD11d/CD18), $\alpha_L\beta_2$ (CD11a/CD18, LFA-1), $\alpha_M\beta_2$ (CD11b/CD18, complement receptor 3 (CR3), Mac1) and $\alpha_X\beta_2$ (CD11c/CD18, CR4, p150/195) integrins, respectively (*Arnaout, 1990*; *Mazzone and Ricevuti, 1995*; *Sanchez-Madrid, 1983*; *Tan, 2012*; *Van der Vieren et al., 1995*). The $\beta_2$ integrins have specialized roles in immune and inflammatory responses and, like other integrins, employ a two-step mechanism of bi-directional signal transmission between the interior of cells and the extracellular milieu (*Anthis and Campbell, 2011*; *Tan, 2012*). Upon activation by various intracellular signals, the inside-out signaling is initiated through rearrangement of the integrin molecule from an inactive (bent, closed, resting, low-affinity) conformation to an active (extended, open, high-affinity) conformation. Subsequent ligand binding triggers outside-in signaling of the extended integrins through activation of Src family tyrosine kinases (*Jakus et al., 2007*; *Mócsai et al., 2002*; *2010*; *Schymeinsky et al., 2007*). Src kinases phosphorylate tyrosine residues within the so-called immunoreceptor tyrosine-based activation motif (ITAM), on the cytoplasmic face of ITAM-containing transmembrane adaptor

**eLife digest** The outer surfaces of animal cells are coated with proteins, including many that are able to sense signals from the environment. The integrins are one such group of proteins. Particular ions or small molecules – collectively known as ligands – can bind to these proteins and activate cascades of signaling events inside the cell.

An integrin called complement receptor 3 (CR3) resides on the surface of many immune cells. CR3 binds to molecules found on the surface of bacteria, and prompts the immune cell to engulf and destroy the bacteria. The ligands bind to a region of CR3 called the I-domain, and it is thought that this domain is only able to accept ligands once the integrin protein has adopted an active form.

*Bordetella pertussis* – the bacterium that causes a disease called whooping cough – subverts the immune defenses of the host. *B. pertussis* produces a toxin known as adenylate cyclase toxin (CyaA) that binds to CR3 in order to penetrate the immune cell and stop immune responses from being activated. However, it is not clear how CyaA is able to bind to CR3 without activating the signaling cascades.

Here, Osicka et al. used biochemical techniques to address this question. The experiments reveal that CyaA mostly binds to an inactive form of CR3 through a unique site outside of the I-domain. It enables the toxin to use the integrin without triggering an immune response. Furthermore, the experiments show how CyaA prevents ligand signaling via CR3 proteins to allow *B. pertussis* to shut down the host's first line of defense against infection.

Osicka et al.'s findings show how CyaA evades the host's immune system and highlight the central role played by this toxin in *B. pertussis* infections. In the future, these findings could inform efforts to produce more effective vaccines against whooping cough.

proteins, such as DAP12 or the FcR γ-chain (FcRγ) (*Jakus et al., 2007*; *Mócsai et al., 2006*; *2010*; *Schymeinsky et al., 2007*). These serve as docking sites for the tandem phosphotyrosine-binding Src homology 2 (SH2) domains of the non-receptor spleen tyrosine kinase (Syk) (*Jakus et al., 2007*; *Mócsai et al., 2006*; *2010*; *Schymeinsky et al., 2007*). Recruitment and activation of Syk then leads to assembly of a multi-protein signaling complex that contains cytosolic Syk-binding molecules and initiates further downstream signaling, ultimately triggering various cellular responses that play a central role in the innate immune defense to infection (*Mócsai et al., 2010*).

The $\beta_2$ integrin complement receptor 3 (CR3) is used as receptor by the 1706 residue-long RTX (Repeats in ToXin) adenylate cyclase toxin-hemolysin (CyaA, ACT, or AC-Hly) of *Bordetella pertussis*, which plays a crucial role in virulence and immune evasion of the whooping cough agent (*Goodwin and Weiss, 1990*; *Guermonprez et al., 2001*; *Khelef et al., 1992*; *Vojtova et al., 2006*; *Weiss et al., 1984*). Upon binding to CR3, CyaA penetrates myeloid phagocytes and paralyzes their bactericidal functions by catalyzing uncontrolled conversion of cytosolic ATP to the key signaling molecule cAMP (*Confer and Eaton, 1982*; *Guermonprez et al., 2001*). The toxin also hijacks maturation and proinflammatory signaling of CR3-expressing dendritic cells and it likely subverts antigen presentation and induction of adaptive T cell immune responses to bacterial infection by intraepithelial dendritic cells of host respiratory mucosa (*Bagley et al., 2002*; *Boschwitz et al., 1997*; *Boyd et al., 2005*; *Fedele et al., 2010*; *Hickey et al., 2008*; *Njamkepo et al., 2000*; *Ross et al., 2004*; *Spensieri et al., 2006*).

Here we reveal that CyaA acts as a unique type of ligand of the I-domain-containing integrin CR3, preferentially recognizing an inactive state of the integrin through interaction at a novel binding site located outside of the I-domain. CyaA thereby avoids activation of downstream signaling of the engaged CR3 *via* the key signaling kinase Syk in human monocytes. Furthermore, we show that CyaA-catalyzed elevation of cAMP effectively blocked the iC3b opsonin-elicited activation of the CR3-Syk signaling pathway in human monocytes.

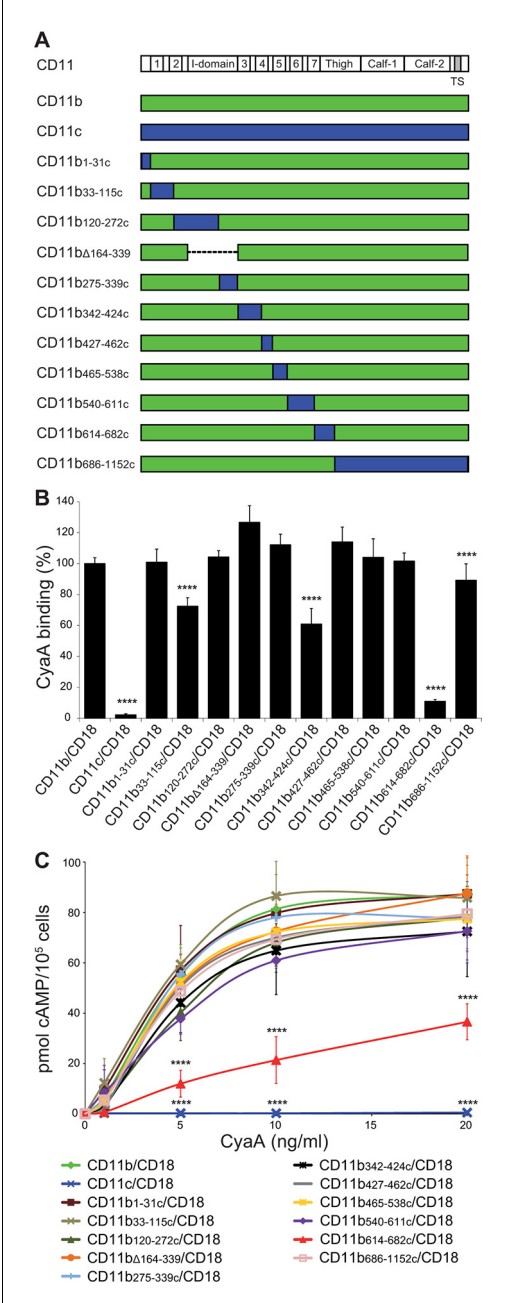

**Figure 1.** Residues 614-682 of CD11b are crucial for CyaA binding and penetration into cells. (**A**) The CD11 subunits of $\beta_2$ integrins consist of a long N-terminal extracellular domain, a single-pass transmembrane segment (TS) and a short C-terminal cytoplasmic tail, respectively. The N-terminal part of the extracellular domain harbors seven $\beta$-sheet repeats (numbers in boxes), forming a $\beta$-propeller domain, which is followed by the thigh, calf-1 and calf-2 domains. The I-domain segment, inserted between repeats 2 and 3 of the $\beta$-propeller domain, plays a critical role in interaction of the I-domain-containing integrins with their endogenous ligands. To map the CyaA binding site on the CD11b subunit, segments of CD11b (green) were systematically replaced with their CD11c counterparts (blue). In the CD11b$_{\Delta164-339}$ molecule, the entire I domain of CD11b was deleted. (**B**) $2\times10^5$ CHO cells expressing integrin molecules were incubated with 2 $\mu$g/ml of CyaA-biotin, the surface-bound toxin was labeled with streptavidin-PE and the cells were analyzed by flow cytometry. CyaA binding was expressed as percentage of toxin binding to CHO cells expressing the native form of CD11b/CD18. Each bar represents the mean value with SD of at least five independent experiments performed in duplicate or triplicate. Significantly reduced binding of CyaA to mutant integrins in comparison with intact CD11b/CD18 is indicated (****, p<0.0001; ANOVA). (**C**) $1\times10^5$ CHO cells expressing integrin molecules were incubated with various concentrations of CyaA and the amounts of

*Figure 1 continued on next page*

*Figure 1 continued*

accumulated cAMP were determined in cell lysates by ELISA. Each point represents the mean value ± SD of at least seven determinations from at least three independent experiments. Significant differences between mean values of cAMP intoxication of cells expressing intact CD11b/CD18 and mutant integrins are shown (****, p<0.0001; ANOVA).

The following figure supplements are available for figure 1:

**Figure supplement 1.** Expression of the CD18 subunit on the surface of CHO cells.

**Figure supplement 2.** Binding of mAbs and CyaA to CHO cells expressing CD11b/CD18, CD11c/CD18 and the CD11b-CD11c/CD18 chimeras.

**Figure supplement 3.** CyaA recognizes CD11b only in the heterodimeric complex with CD18.

## Results

### CyaA binds CR3 outside of its ligand-binding I-domain

It was previously shown that Chinese hamster ovary (CHO) cells expressing human CR3 can be used as a suitable model for studying the interaction of CyaA with CR3 (*Guermonprez et al., 2001*). Indeed, CR3 (CD11b/CD18) expressed by CHO cells allowed the binding and cAMP-elevating (cyto-toxic) activities of CyaA, while the highly homologous CR4 (CD11c/CD18) was unable to bind CyaA despite sharing the same $\beta_2$ (CD18) subunit with CR3 (*Guermonprez et al., 2001*). Therefore, to delineate the CyaA binding site(s) on CR3, we performed swapping of the homologous alpha chain segments (CD11b and CD11c) of CR3 and CR4 (*Figure 1A*). First, a CHO cell line stably expressing the CD18 subunit (CHO-CD18) was established (*Figure 1—figure supplement 1*). This was next used for generating stable cell lines that expressed similar quantities of intact CD11b (CHO-CD11b/CD18), CD11c (CHO-CD11c/CD18), or of the chimeric CD11b-CD11c heterodimeric complexes on the cell surface (*Figure 1—figure supplement 2*). The capacity of such cells to bind CyaA was then assessed by flow cytometry and the susceptibility to CyaA penetration and enzymatic intoxication by the delivered AC domain of the toxin was measured as intracellular accumulation of cAMP. As shown in *Figure 1B*, replacement of the segment containing residues 614 to 682 (segment 614-682) of CD11b by the corresponding portion of CD11c caused a sharp, about ten-fold decrease of CyaA binding to the CHO-CD11b$_{614-682c}$/CD18 transfectants, as compared to cells expressing intact CR3 (*Figure 1B*). This resulted in accordingly reduced toxin penetration and cAMP accumulation in cells (*Figure 1C*). CyaA binding was also partially reduced on cells expressing CR3 chimeras having the residues 33 to 115 (72 ± 6%), 342 to 424 (61 ± 10%), or 686 to 1152 (89 ± 11%) of CD11b replaced by the corresponding segments of CD11c, respectively (*Figure 1B*). No reduction of CyaA binding was observed on cells expressing the remaining CD11b-CD11c/CD18 chimeras (*Figure 1B*), which all accumulated comparable levels of cAMP as the cells expressing intact CR3 (*Figure 1C*). Surprisingly, CyaA binding and subsequent intoxication of cells by cAMP were neither reduced by swapping of the homologous I-domain regions between CD11b and CD11c (CD11b$_{120-272c}$ and CD11b$_{275-339c}$), nor upon removal of the entire I-domain of CD11b (CD11b$_{\Delta164-339}$) (*Figure 1B,C*). This demonstrates that CyaA does not bind CR3 through the I-domain and rather recognizes the segment containing residues 614 to 682 that encompasses the C-terminal end of the last repeat of the β-propeller domain and the N-terminal portion of the thigh domain of CD11b. This finding is consistent with the sequence alignment by the Jotun Hein method (Lasergene software, DNASTAR) showing that the 614-682 segments of CD11b and CD11c are only 41% identical at the amino acid level, compared to the entire CD11b and CD11c subunits that exhibit an overall 61% sequence identity. Since, however, some other segments of CD11b appeared to be important for full toxin binding to CR3 as well, these results suggest that CyaA likely binds the integrin through a multivalent interaction, making contacts with several integrin segments. Moreover, CyaA bound CD11b only when this was associated with CD18 and not when the CD11b subunit was expressed alone on the surface of CHO-CD11b cells (*Figure 1—figure supplement 3*).

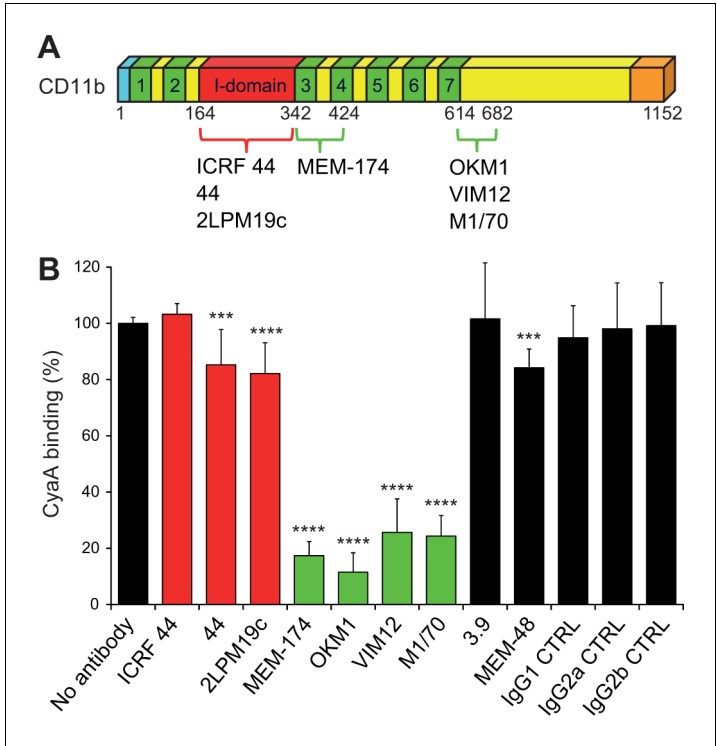

**Figure 2.** Antibodies recognizing the same segments of CD11b as CyaA block its binding to CR3. (**A**) Schematic representation showing the binding segments of a set of mAbs in the CD11b subunit of CR3, which were mapped by flow cytometry. The ICRF 44, 44 and 2LPM19c mAbs recognize the I-domain of CD11b, the major ligand binding site of CR3. The MEM-174, or OKM1, VIM12 and M1/70 mAbs target amino acid segments 342-424 or 614-682 of CD11b, respectively, which are important for CyaA binding. (**B**) $2\times10^5$ CHO-CD11b/CD18 cells were preincubated without or with saturating concentrations of different mAbs and then incubated with 2 μg/ml of CyaA-biotin. Surface-bound CyaA-biotin was labeled with streptavidin-PE and the cells were analyzed by flow cytometry. CyaA binding was expressed as percentage of toxin binding to CHO-CD11b/CD18 cells treated without mAb. Each bar represents the mean value with SD of at least eight determinations from at least three independent experiments. Significant differences between mean values of CyaA binding to mAb-untreated cells and cells treated with different mAbs are indicated (***, $p < 0.001$; ****, $p < 0.0001$; ANOVA). 3.9, CD11c-specific mAb; MEM-48, CD18-specific mAb; IgG1, IgG2a and IgG2b, isotype control mAbs.

The following figure supplements are available for figure 2:

**Figure supplement 1.** Flow cytometry profiles of anti-CD11b mAbs on CHO cells expressing the integrin CD11b/CD18 and its mutant variants.

**Figure supplement 2.** Anti-CD11b antibodies do not activate CR3.

---

To exclude that the reduction of CyaA binding may have resulted from a structural alteration of the integrin chimeras, we next mapped the CyaA binding site on intact CR3 by performing competition experiments with a set of CD11b-specific monoclonal antibodies (mAbs; *Figure 2A* and *Figure 2—figure supplement 1*) that do not activate CR3 (*Figure 2—figure supplement 2*). The I-domain-recognizing mAbs ICRF 44, 44 and 2LPM19c (*Figure 2A* and *Figure 2—figure supplement 1*), the CD11c-specific mAb 3.9, the CD18-specific mAb MEM-48 and the isotype control mAbs, all failed to substantially reduce CyaA binding to CR3 (*Figure 2B*). In contrast, CyaA was strongly outcompeted from binding to intact CR3 by the excess of OKM1, VIM12, or M1/70 mAbs (*Figure 2B*) that recognize epitopes located between the residues 614 to 682 of CD11b (*Figure 2A* and *Figure 2—figure supplement 1*). This strongly supports the conclusion reached by the CD11b-CD11c domain swapping experiments that the segment encompassing residues 614 to 682 of CD11b is specifically involved in binding of CyaA. In addition, CyaA binding to CR3 was strongly inhibited by

the MEM-174 mAb (*Figure 2B*) that recognizes the CD11b segment comprised between residues 342 to 424 (*Figure 2A* and *Figure 2—figure supplement 1*). This also concurs with the results of the segment swapping experiments, showing that the segment 342-424 along with some other segments of CD11b may also be contributing to CyaA binding to CR3.

To confirm that the 614-682 segment of CD11b is the principal binding site of CyaA, we transferred it into the context of the CD11c subunit of CR4 (*Figure 3A*). Since mAbs recognizing CD11b and CD11c with equal efficacy are not available, a fluorescent YFP tag was added to the C-termini of CD11b, CD11c and CD11c$_{614-682b}$ molecules (*Figure 3A*). This allowed cytometric selection of stably transfected CHO clones expressing similar quantities of the integrins on the cell surface (*Figure 3—figure supplement 1*). As shown in *Figure 3B*, the CD11c$_{614-682b}$-YFP/CD18 chimera bound different concentrations of CyaA with substantially higher efficacy (on average 24-fold higher) than the intact CD11c-YFP/CD18 complex and the chimera exhibited only a slightly lower efficacy of toxin binding (on average 1.7-fold lower) than the CD11b-YFP/CD18 integrin itself (*Figure 3B*). Consistently, the CHO-CD11c$_{614-682b}$-YFP/CD18 cells were efficiently penetrated by CyaA and accumulated high levels of cAMP (*Figure 3C*). The segment encompassing residues 614 to 682 of CD11b, hence, constitutes an autonomous CyaA-binding structure that enables toxin penetration into cells even when located in the context of the CD11c/CD18 integrin molecule.

## CyaA recognizes its binding site on CD11b through electrostatic interaction

To analyze the binding interaction of CyaA with CR3 in molecular detail, we employed a set of *in silico* approaches to predict the key residues involved. First, the known 3D structure of the homologous CR4 integrin (*Xie et al., 2010*) was used as a template for construction of a homology model of the 3D structure of CR3. In parallel, the structure of the CR3 binding site located within residues 1166 to 1287 (segment 1166-1287) of CyaA (*El-Azami-El-Idrissi et al., 2003*) was predicted *ab initio*, using I-TASSER (*Roy et al., 2010*). Flexible side chain docking of CR3 and of the segment 1166-1287 of CyaA by the ClusPro server (*Comeau et al., 2007*) was next used to predict the most probable interacting residues of the two proteins (*Figure 4A,B*). Second, the α subunits of all four β$_2$ integrins were aligned by the Jotun Hein method (Lasergene software, DNASTAR) and residues that are present at a given position only in the 614-682 segment of CD11b were considered as potentially involved in CyaA binding. Finally, the theoretical isoelectric points of the interacting segments (3.76 for 1166-1287 segment of CyaA and 8.27 for 614-682 segment of CD11b) were taken into account. The sum of these considerations suggested that the negatively charged residues of the 1166-1287 segment of CyaA might be involved in the interaction with positively charged residues within the 614-682 segment of CD11b.

Therefore, the predicted negatively charged glutamate and aspartate residues of the 1166-1287 segment of CyaA were substituted by alanine. As shown in *Figure 4C,D*, the CyaA$_{D1193A+D1194A+E1195A}$ and CyaA$_{E1232+D1234A}$ constructs exhibited a strongly reduced capacity to bind and penetrate CR3-expessing CHO cells. Further, combinations of alanine substitutions were also introduced in place of charged and hydrophilic residues within the 614-682 segment of CD11b and six mutant CD11b molecules were stably expressed on the surface of CHO-CD18 cells (*Figure 4—figure supplement 1*). No statistically significant drop of CyaA binding (*Figure 4E*) and cAMP elevation (*Figure 4F*) was observed in the cells expressing CR3 variants with the combinations of substitutions R619A+K621A, E625A+N627A+R629A, K644A+K646A, or E650A+R652A within the 614-682 segment of CD11b, respectively. In contrast, alanine substitutions of the three positively charged arginine residues at positions 662, 664 and 666 of CD11b almost completely abolished binding of CyaA to the CR3 mutant variant and a significant reduction of CyaA binding was also observed with cells expressing the CD11b$_{K659+T661A}$/CD18 construct (*Figure 4E*). As shown in *Figure 4—figure supplement 2*, this loss or reduction of CyaA binding was not due to an alteration of the structure of the CyaA binding site within the mutant integrins, as these bound the CyaA-blocking mAbs MEM-174, OKM1 and VIM12 to the same extent as the intact CR3. The only exception was the CyaA-blocking mAb M1/70, which exhibited reduced binding to both mutant integrins, similarly as CyaA. This indicates that the M1/70 mAb and CyaA may recognize the same epitope within the 614-682 segment of CD11b. In comparison to CHO cells expressing intact CR3, CyaA produced importantly lower intracellular cAMP levels in cells expressing CD11b$_{R662A+R664A+R666A}$/CD18 (*Figure 4F*). Taken

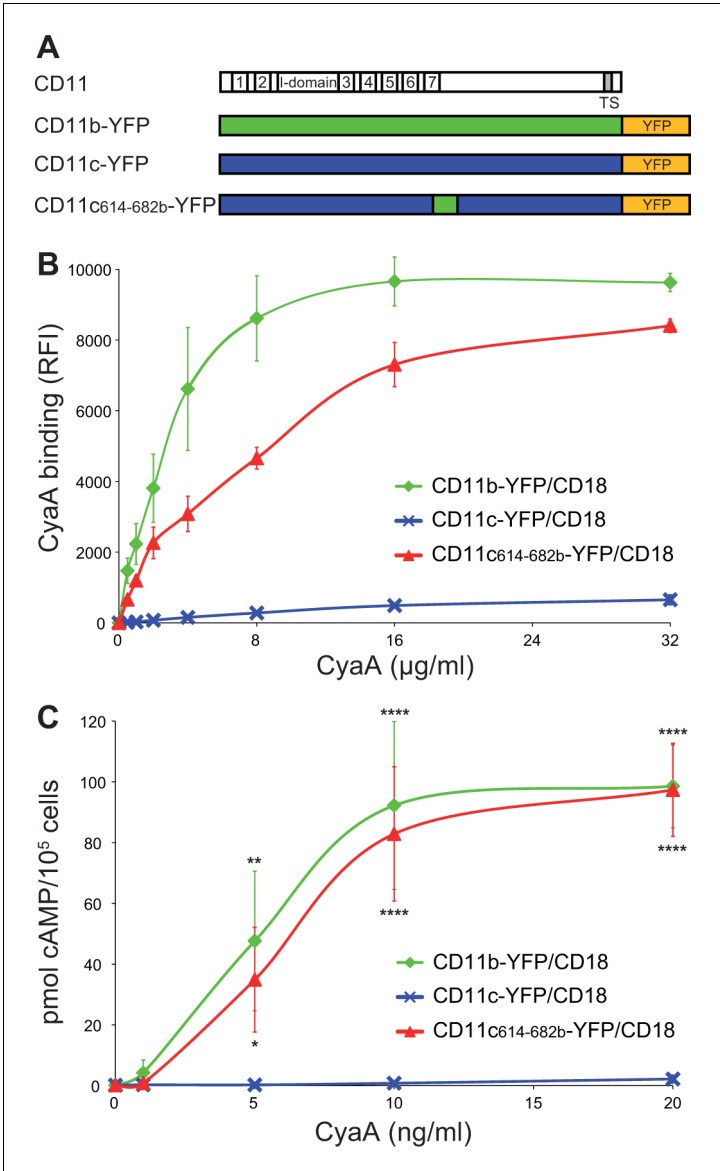

**Figure 3.** Residues 614 to 682 of CD11b confer CyaA binding to CD11c. (**A**) Schematic representation of the CD11c subunit (blue) with residues 614–682 replaced by the corresponding residues of CD11b (green). All CD11 molecules were C-terminally fused to yellow fluorescent protein (YFP) to allow quantification of expression. (**B**) $2 \times 10^5$ CHO cells expressing integrin molecules were incubated with different concentrations of CyaA-biotin and the surface-bound toxin was labeled with streptavidin-PE. The cells were analyzed by flow cytometry and mean fluorescence intensities of CyaA binding were plotted against the concentrations of CyaA. Each point represents the mean value ± SD of two independent experiments performed in triplicate. CyaA binding to cells expressing CD11c$_{614-682b}$-YFP/CD18, or CD11b-YFP/CD18 was at all measured CyaA concentrations significantly higher in comparison with cells expressing intact CD11c-YFP/CD18 ($p<0.0001$; ANOVA). RFI, relative fluorescence intensity. (**C**) $1 \times 10^5$ CHO cells expressing integrin molecules were incubated with various concentrations of CyaA and the amounts of accumulated cAMP were determined in cell lysates by ELISA. Each point represents the mean value ± SD of two independent experiments performed in duplicate. Significant differences between mean values of cAMP intoxication of cells expressing CD11c$_{614-682b}$-YFP/CD18, or CD11b-YFP/CD18 and intact CD11c-YFP/CD18 are shown (*, $p<0.05$; **, $p<0.01$; ****, $p<0.0001$; ANOVA).

The following figure supplement is available for figure 3:

**Figure supplement 1.** Expression of integrin variants fused with a fluorescent YFP protein on the surface of CHO cells and binding of CyaA to transfected cells.

together, our mutagenesis experiments confirmed the computational predictions and identified the residues involved in CD11b recognition by CyaA.

## CyaA preferentially binds an inactive, low-affinity conformation of CR3

Since endogenous ligands preferentially bind the I-domain of activated CR3, the unique location of the CyaA binding site prompted us to test whether CyaA differentiates between the inactive (bent) and active (extended) conformation of CR3. For this experiment, a leukocyte-enriched fraction was isolated from fresh human blood and treated (or not) with phorbol 12-myristate 13-acetate (PMA) that is a known agonist activating CR3 (*Diamond and Springer, 1993*). The cells were subsequently stained with mAbs and exposed to different concentrations of CyaA, before being analyzed by flow cytometry. Monocytes were gated based on light-scatter characteristics and expression of CD14 and used for assessment of CyaA binding. As shown in *Figure 5A,B*, PMA-stimulated monocytes expressed the same amounts of CR3 as untreated monocytes (detected with OKM1 mAb), but bound the toxin with a significantly lower efficiency. This was clearly due to conformational rearrangement of CR3 on PMA-activated monocytes, as the activated cells bound approximately two-fold more of the activation-reporting mAb MEM-148 (*Drbal et al., 2001*) than the non-activated monocytes (*Figure 5A*). The monocytes pretreated with PMA then accumulated substantially lower intracellular levels of cAMP upon exposure to CyaA than the non-activated cells (*Figure 5C*). Similarly, activation of CR3 by pretreatment of cells with $Mn^{2+}$ ions (*Altieri, 1991*; *Li et al., 2013*) reduced toxin binding and cAMP accumulation in monocytes (*Figure 5—figure supplement 1*).

To further analyze the impact of CR3 conformation on CyaA binding, a soluble secreted form of the CR3 ectodomain complex (sCR3) was purified from supernatants of CHO cultures by immunoaffinity chromatography on MEM-174 mAb-Sepharose beads. The bent and extended conformers of sCR3 were resolved by size exclusion chromatography (*Figure 5—figure supplement 2A*), their conformation was confirmed by electron microscopy (*Figure 5—figure supplement 2B,C*) and the two sCR3 conformers were immobilized onto a surface plasmon resonance (SPR) sensor chip. Coupling densities and random orientation of sCR3 on the chip was verified by passage of the control mAb MEM-48 that recognizes both integrin conformations (*Figure 5D*). As shown in *Figure 5E*, the immobilization procedure preserved the conformation of the integrin molecules, since the integrin activation-reporting mAb MEM-148 bound only the immobilized extended sCR3 and not its bent form.

Since aggregation and unspecific binding to the SPR chip surface of intact CyaA was observed at the required toxin concentrations, we used a CyaAΔH construct, which lacks the hydrophobic pore-forming domain and is soluble even at high concentrations (*Šebo and Ladant, 1993*). Real-time SPR measurements of CyaAΔH interaction with the immobilized sCR3 conformers exhibited typical association and dissociation curves (*Figure 5F,G*). Analysis of the binding curves by a bivalent analyte model then revealed that the association rate was higher and the dissociation rate was lower for the complex of the bent sCR3 conformer with CyaAΔH ($K_d = 2.1 \times 10^{-7}$ M) than for the complex of the extended sCR3 conformer with CyaAΔH ($K_d = 6.4 \times 10^{-7}$ M) (*Table 1*). This demonstrates that the bent CR3 conformer bound the toxin with a higher affinity than the extended conformer, confirming the data obtained by binding experiments on PMA- or $Mn^{2+}$-activated primary monocytes.

## CyaA binding to CR3 does not initiate downstream signaling

Binding of endogenous ligands to the I-domain of CR3 is known to trigger activation of Syk. This initiates downstream signaling and provokes multiple cellular activation responses, including complement-mediated opsonophagocytosis of bacteria (*Shi et al., 2006*). We hence examined if the I-domain-independent interaction of CyaA with CR3 activates Syk in human monocytic cells. THP-1 monocytes, which bind CyaA in a CR3-dependent and saturable manner (*Figure 6—figure supplement 1*), were exposed to different concentrations of CyaA over a range of incubation times. Phosphorylated Syk was next immunoprecipitated from cell lysates with anti-phosphotyrosine mAb and was detected by anti-Syk mAb in immunoblots. As a positive control, the CR3-Syk signaling pathway was activated by iC3b-opsonized zymosan particles (*Shi et al., 2006*). As shown in *Figure 6A*, Syk phosphorylation on tyrosine residues was observed after 30 min of incubation of THP-1 cells with iC3b-opsonized zymosan. No phosphorylation of Syk was, however, detected following exposure of THP-1 cells for up to 60 min to 30 ng/ml CyaA, a concentration close to physiological toxin levels (*Eby et al., 2013*). Similarly, Syk activation was not observed when THP-1 cells were incubated with

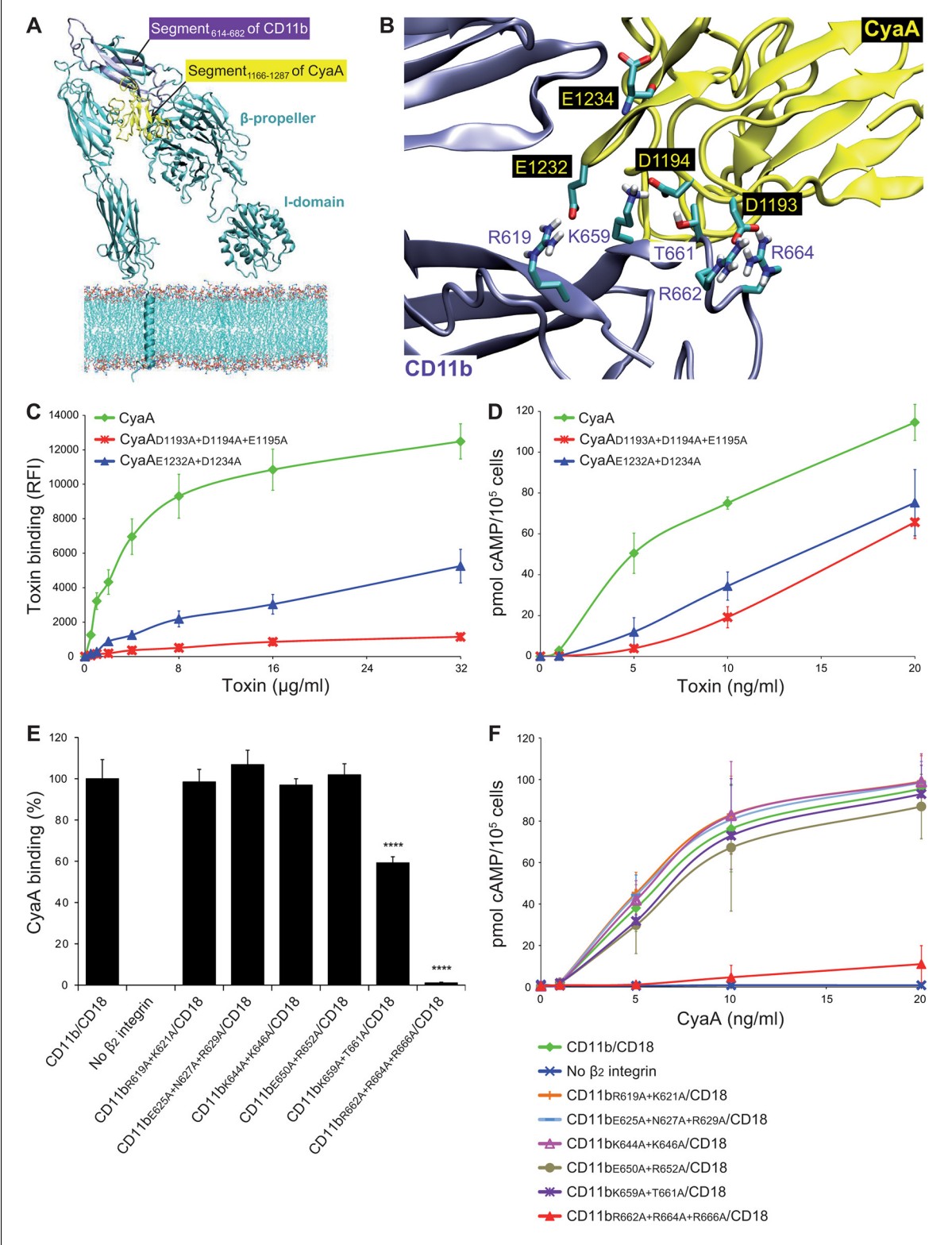

**Figure 4.** Electrostatic interaction of oppositely charged residues underlies CyaA binding to CR3. (A) 3D structure of CR3 was modeled by homology onto the known 3D structure of CR4. The structure of the CD11b binding site within the segment 1166-1287 of CyaA was predicted using I-TASSER. For clarity, only the CD11b subunit is shown. (B) To identify interacting residues, a flexible side chain docking of the segment 1166-1287 of CyaA to CR3 was performed using the ClusPro server. (C) Different concentrations of Dy647-labeled intact CyaA or its variants with point mutations in the CD11b binding site of the toxin were incubated with 2x10⁵ CHO cells expressing intact CR3 and the cells were analyzed by flow cytometry. Mean fluorescence

*Figure 4 continued on next page*

*Figure 4 continued*

intensities of binding of intact CyaA or its variants were plotted against the toxin concentrations. Each point represents the mean value ± SD of four independent experiments. Binding of CyaA mutant variants to cells was at all measured concentrations significantly lower than binding of intact CyaA (p<0.0001; ANOVA). RFI, relative fluorescence intensity. (D) Different concentrations of intact CyaA and its mutant variants were incubated with $1\times10^5$ CHO cells expressing intact CR3 and intracellular levels of cAMP were determined by ELISA. Each point represents the mean value ± SD of two independent experiments performed in triplicate. Intoxication of cells by CyaA mutant variants was at all measured concentrations significantly lower than intoxication of cells by intact CyaA (p<0.0001; for CyaA$_{E1232+D1234A}$ at 20 ng/ml p<0.001; ANOVA). (E) $2\times10^5$ CHO cells expressing integrin molecules were incubated with 2 µg/ml of CyaA-biotin, the surface-bound toxin was labeled with streptavidin-PE and the cells were analyzed by flow cytometry. CyaA binding was expressed as percentage of toxin binding to CHO cells expressing the native form of CD11b/CD18. Each bar represents the mean value with SD of two independent experiments performed in duplicate. Significantly reduced binding of CyaA to mutant integrins in comparison with intact CD11b/CD18 is indicated (****, p<0.0001; ANOVA). (F) $1\times10^5$ CHO cells expressing different integrin molecules were incubated with various concentrations of CyaA and the amounts of accumulated cAMP were determined in cell lysates by ELISA. Each point represents the mean value ± SD of six independent experiments. Intoxication of cells expressing CD11b$_{R662A+R664A+R666A}$/CD18 or no β₂ integrin was in concentrations ranging from 5 to 20 ng/ml of CyaA significantly lower than intoxication of cells expressing intact CD11b/CD18 (p<0.0001; ANOVA).

The following figure supplements are available for figure 4:

**Figure supplement 1.** Expression of CD11b/CD18 and of its mutant variants on the surface of CHO cells and binding of CyaA to transfected cells.

**Figure supplement 2.** Staining of CHO cells expressing intact CD11b/CD18 and of its two mutant variants with the four mAbs that block the binding of CyaA.

the enzymatically inactive CyaA-AC⁻ toxoid (*Figure 6B*). Neither was Syk phosphorylation triggered by THP-1 cell exposure to concentrations of up to 3 µg/ml of active CyaA toxin (*Figure 6C*), or of its CyaA-AC⁻ toxoid (*Figure 6D*) for 15 or 30 min, respectively. Hence, in contrast to iC3b, which binds through the I-domain of CR3 and triggers activation of Syk (*Shi et al., 2006*), the I-domain-independent mode of CyaA binding did not trigger any CR3 signaling through Syk pathway activation.

## CyaA-generated cAMP effectively blocks the CR3-Syk signaling pathway

We further investigated whether intracellular cAMP signaling produced by the cell-invading CyaA toxin would interfere with the outside-in signaling of CR3. As shown in *Figure 7A*, Syk was efficiently activated by iC3b-opsonized zymosan in THP-1 cells that were pre-treated with buffer, or with 300 ng/ml of the enzymatically inactive CyaA-AC⁻ toxoid. However, iC3b-coated particles failed to elicit any Syk phosphorylation in THP-1 cells that were pre-incubated with 300 ng/ml of CyaA (*Figure 7A*). Control experiments demonstrated that CyaA and CyaA-AC⁻ bound THP-1 cells with the same efficacy (*Figure 7—figure supplement 1*) and that toxin binding did not alter the level of CR3 expression on the cell surface (*Figure 7—figure supplement 2*). Nevertheless, a partial reduction of binding of the iC3b-coated particles to THP-1 cells preincubated with CyaA was observed, as compared to cells preincubated with buffer alone or with the CyaA-AC⁻ toxoid (*Figure 7—figure supplement 3*). Hence, the cAMP signaling action of CyaA concurrently prevented Syk activation by CR3-bound iC3b as well as it provoked a decrease of the iC3b opsonin binding.

Importantly, when Syk was pre-activated by incubation of THP-1 cells with iC3b-opsonized zymosan, the subsequent addition of CyaA, but not of CyaA-AC⁻, provoked a significant drop of tyrosine phosphorylation of Syk (*Figure 7B*). Control experiments showed that preincubation of THP-1 cells with iC3b-opsonized zymosan did not reduce binding of CyaA or CyaA-AC⁻, respectively (*Figure 7—figure supplement 4*). Moreover, the impairment of Syk phosphorylation upon CyaA treatment was not due to decreased cell viability (*Figure 7—figure supplement 5*). It can, hence, be concluded that Syk inactivation was due to signaling of the cAMP produced in cells by the penetrating CyaA toxin. Similar results were obtained when the same experiments were performed with primary human monocytes isolated from leukopacks of healthy donors (*Figure 7—figure supplement 6*). Taken together, our data show that the cAMP signaling of CyaA decreases the capacity of CR3 to bind the iC3b opsonin, prevents activation of Syk by iC3b-engaged CR3 and inactivates also Syk that has already been activated by iC3b/CR3.

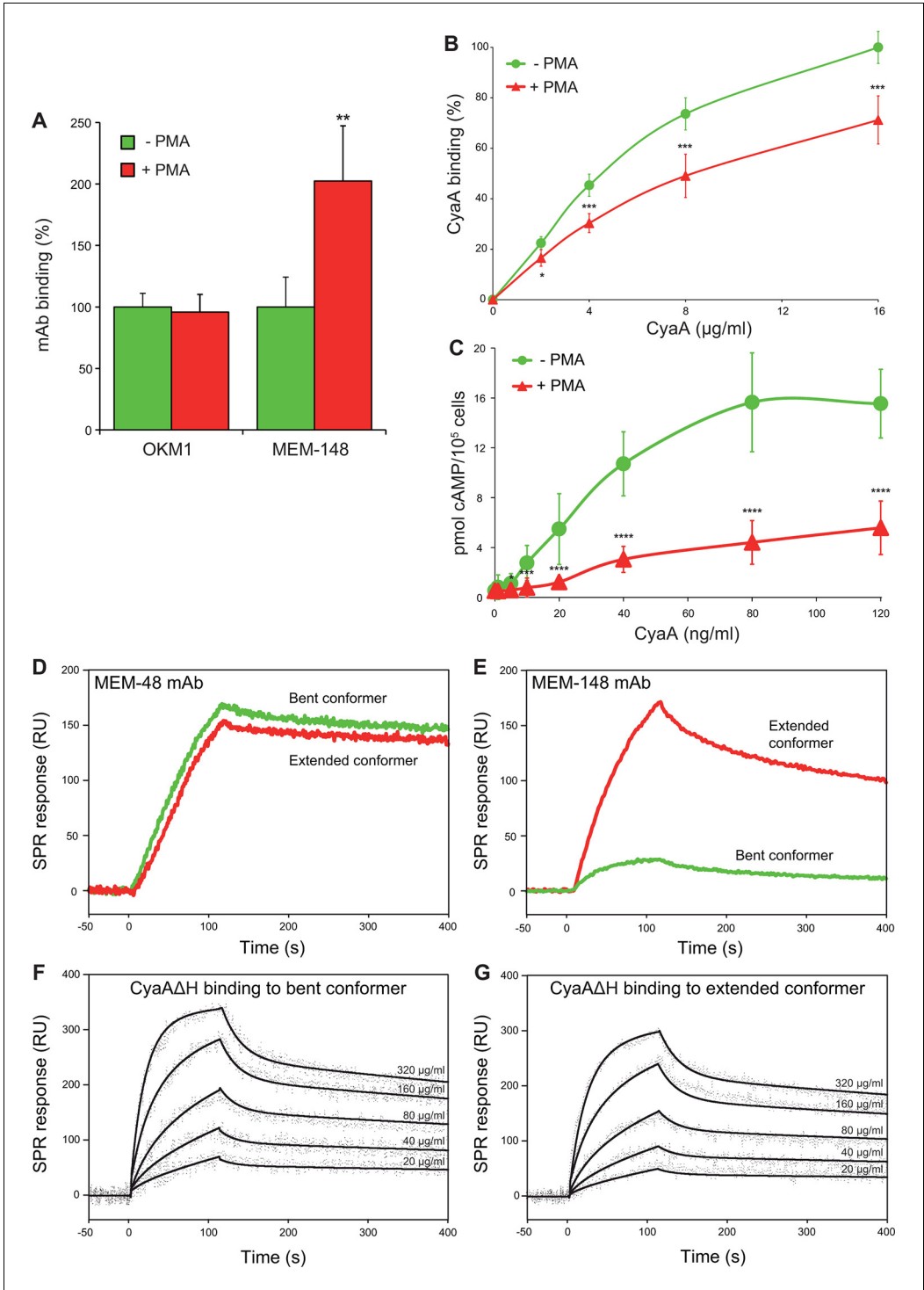

**Figure 5.** CyaA preferentially recognizes the inactive (bent) conformation of CR3. (**A, B**) A leukocyte-enriched fraction prepared from fresh whole blood was treated without or with 100 nM PMA to activate CR3. The cells were promptly stained with the OKM1 mAb recognizing both CR3 conformations, or with the MEM-148 mAb recognizing the extended integrin conformation (**A**), or with different concentrations of Dy647-labeled CyaA (**B**) in a combination with anti-CD14 mAb. After 2 min, cells were analyzed by flow cytometry, monocytes were gated based on light-scatter characteristics and expression of CD14 and used for calculation of mAbs and CyaA binding. Binding of mAbs, or CyaA (at the highest tested concentration) to PMA-untreated cells was taken as 100%. Each value represents the mean with SD of three independent experiments performed in duplicate using three different donors. Significant differences between mean values of mAbs or CyaA binding to cells treated with buffer alone and cells treated with PMA are shown (*, p<0.05; **, p< 0.01; ***, p<0.001; Student's t-test). (**C**) 1x10⁵ human primary monocytes were pretreated without or with 100 nM PMA and incubated with indicated concentrations of CyaA. The amounts of accumulated cAMP were determined in cell lysates by ELISA. Each point represents

*Figure 5 continued on next page*

*Figure 5 continued*

the mean value ± SD of seven independent experiments performed in duplicate using cells of seven different donors. Significant differences between mean values of cAMP intoxication of monocytes incubated in the absence and in the presence of PMA are shown (*, p<0.05; ***, p<0.001; ****, p<0.0001; Student's t-test). (D-G) The bent and extended conformers of sCR3 were immobilized to a Bio-Rad ProteOn XPR36 GLC sensor chip and the MEM-48 mAb recognizing both conformations of sCR3 (D), or the MEM-148 mAb recognizing the extended integrin conformation (E) were used as controls. To analyze the interaction between the bent (F), or extended (G) conformation of sCR3 and the toxin, CyaAΔH was passed over the chip surface at concentrations of 20, 40, 80, 160 and 320 µg/ml. The data were analyzed by global fitting of the response curves using a bivalent analyte model and calculated kinetic parameters are given in *Table 1*.

The following figure supplements are available for figure 5:

**Figure supplement 1.** CyaA preferentially recognizes the inactive (bent) conformation of CR3.

**Figure supplement 2.** Isolation and characterization of the bent and extended conformers of sCR3.

## Discussion

We show here that CyaA hijacks the complement receptor 3 of phagocytes by subversively binding to its bent, inactive conformation by binding to a unique site located outside of the I-domain. This novel type of CR3 interaction then enables rapid toxin penetration into phagocytic cells and elevation of cytosolic cAMP concentration, thus ablating the pro-phagocytic signaling of CR3 via Syk (*Figure 8*). This unprecedented mode of binding and action differentiates CyaA from all other known ligands of the I-domain-containing integrins.

While the I-domain is essential for binding of ligands to I-domain-containing integrins, the CyaA toxin binds CR3 independently of the presence of the I-domain. This possibility has already been suggested by Guermonprez and co-workers, who observed that in contrast to I-domain-dependent CD11b ligands, CyaA binding does not require the presence of $Mg^{2+}$ ions (*Guermonprez et al., 2001*). Indeed, we show that CyaA primarily recognizes the CD11b segment consisting of residues 614 to 682. This encompasses the C-terminal end of the last β-propeller repeat and the N-terminal portion of the thigh domain.

Besides interacting with the I-domain, certain ligands appear to interact to some extent also with the β-propeller domain of alpha subunits of $\beta_2$ integrins (*Li and Zhang, 2003*; *Yalamanchili et al., 2000*). The β-propeller domain consist of seven β-sheet repeats (*Figure 1A*) that are arranged in a torus, or propeller, with each β-sheet representing one blade of the propeller (*Oxvig and Springer, 1998*; *Springer, 1997*; *Xie et al., 2010*). Its fold is most closely related to the β subunit of the trimeric G-protein (*Springer, 1997*) and proper folding of the β-propeller requires association of the α subunit with the $\beta_2$ subunit (*Huang and Springer, 1997*; *Lu et al., 1998*). The β-propeller is followed by the thigh domain, exhibiting a C2-set immunoglobulin fold (*Chothia and Jones, 1997*), and by a genu region (*Xie et al., 2004*; *2010*). The structural rearrangement at the thigh/genu interface is involved in the activation-dependent extension of the integrin α subunit (*Xie et al., 2004*). It was previously suggested that the β-propeller/thigh region segment of CD11b (residues 599 to 679), comprising the principal CyaA binding site, can fold also in the absence of the CD18 subunit, since it was recognized by the OKM1 mAb when the CD11b subunit was expressed alone on COS cells (*Lu et al., 1998*). Indeed, we observed here that OKM1 recognized CD11b that was expressed alone on CHO-CD11b cells (*Figure 1—figure supplement 3A*). However, CyaA failed to bind any significantly to cells expressing the isolated CD11b subunit on their surface (*Figure 1—figure supplement 3C*). In contrast to the OKM1 epitope, hence, the CyaA-binding site within the segment 614-682 of CD11b may acquire the correct structure (fold) only when the CD11b subunit is in complex with CD18. Alternatively, an interaction of CyaA with the CD18 subunit may also be involved in CyaA binding to CR3, even though CD18 alone was unable to bind CyaA on its own (*Figure 1—figure supplement 3C*).

In addition, recognition of additional segments of CD11b appears to contribute to the full capacity of CR3 to bind CyaA. Engagement of these segments likely facilitates or stabilizes the higher-affinity interaction of CyaA with the principal binding site (residues 614 to 682). Toxin interactions at several sites on CD11b would then synergize in bringing about the highly selective interaction of CyaA with the CD11b subunit of CR3. Indeed, we have previously described that an initial low-

**Table 1.** Kinetic parameters of CyaAΔH binding to sCR3 calculated by a bivalent analyte model.

| Ligand[a] | Analyte | $k_{a1}$ [$\times 10^3$ M$^{-1}$s$^{-1}$] | $k_{d1}$ [$\times 10^{-3}$ s$^{-1}$] | $k_{a2}$ [$\times 10^{-4}$ RU$^{-1}$s$^{-1}$] | $k_{d2}$ [$\times 10^{-4}$ s$^{-1}$] |
|---|---|---|---|---|---|
| Bent sCR3 | CyaAΔH | 5.8 ± 0.7 | 1.2 ± 0.2 | 1.3 ± 0.2 | 3.8 ± 0.5 |
| Extended sCR3 | | 2.8 ± 0.2 | 1.8 ± 0.4 | 2.7 ± 0.3 | 5.1 ± 0.6 |

[a]Two independent SPR binding experiments were performed in duplicate and the differences between mean values of CyaAΔH binding to the bent and extended conformation of sCR3 were statistically significant (p<0.01; Student's t-test).

affinity interaction of CyaA with N-linked glycan chains of the C-terminal part of the CD11b subunit is a critical prerequisite for toxin binding to CR3 (*Hasan et al., 2015*; *Morova et al., 2008*). A low-affinity interaction with abundant N-linked glycans on the integrin molecule most likely serves as the initial step of toxin pre-concentration from solution into the two-dimensional space of the cell surface. This increase of local CyaA concentration would then increase the probability of proper positioning of the toxin molecule for its subsequent high affinity interaction with specific amino acid residues of the CR3 glycoprotein. This two-step mode of CyaA-CR3 interaction is supported by the SPR binding curves which fit well using the bivalent analyte model, describing a two-step association process. Compared to a mechanism of direct single step binding from solution, a sequential binding mechanism would, indeed, lower the affinity constraints on interaction of CyaA with the ectodomain of CR3. The stability of the CyaA-CR3 complex may then be further increased by an irreversible insertion of the hydrophobic domain and of the acyl chains of CyaA into the cellular membrane, which most likely accounts for the observed high apparent affinity of toxin binding to CR3-expressing cells (*Guermonprez et al., 2001*).

It was recently proposed that CyaA can specifically bind yet another β₂ integrin on leukocytes, the LFA-1 (CD11a/CD18) complex (*Paccani et al., 2011*). However, this is at odds with the initial observation of Guermonprez and co-workers that CyaA does selectively and with high affinity bind the dendritic cells and macrophages expressing CD11b/CD18, whereas B and T cell lines expressing exclusively the LFA-1 complex were recognized with substantially lower efficacy, if at all (*Guermonprez et al., 2001*). To rule out that this difference in CyaA binding was due to unequal expression levels of the integrins on various cell types, we added a fluorescent YFP tag to the C-terminus of the CD11a subunit and generated a cell line (CHO-CD11a-YFP/CD18) expressing the same quantities of the integrin molecules on the cell surface as found on the CR3-expressing cell line (CHO-CD11b-YFP/CD18) (*Figure 9—figure supplement 1*). As shown in *Figure 9A*, over a range of CyaA concentrations, the CHO-CD11a-YFP/CD18 cells bound about two orders of magnitude less CyaA toxin than the CD11b-YFP/CD18-expressing cells. Moreover, the difference in binding of CyaA to CD11a-YFP/CD18-expressing cells, CD11c-YFP/CD18-expressing cells, or the mock-transfected cells expressing no β₂ integrin at all, was not statistically significant (*Figure 9A*). To exclude the concern that the residual binding of CyaA to CD11a-YFP/CD18 could be due to the presence of the YFP tag, we generated a cell line expressing comparable amounts of intact CD11a/CD18 (*Figure 9—figure supplement 2A*) and demonstrated that CyaA binds these cells as poorly as cells expressing CD11a-YFP/CD18 with the YFP tag (*Figure 9—figure supplement 2B*). In addition, we purified the intact LFA-1 integrin from human peripheral blood mononuclear cells by immunoaffinity chromatography on MEM-25 mAb-Sepharose beads and immobilized it onto an SPR sensor chip. Real-time SPR measurements revealed no CyaAΔH binding to the intact LFA-1 (*Figure 9—figure supplement 3*; cf. CyaAΔH binding to CR3 in *Figure 5F,G*). Finally, cells expressing CD11a-YFP/CD18 were intoxicated by CyaA to equally low cAMP levels as the control cells expressing CD11c-YFP/CD18, or lacking any β₂ integrin, while cAMP intoxication of cells expressing CD11b-YFP/CD18 was over two orders of magnitude higher than that of the control cells (*Figure 9B*). All these data clearly demonstrate that CyaA binds and intoxicates cells expressing LFA-1 with equally low efficacy as cells expressing CR4, or lacking any β₂ integrin at all.

Our findings have numerous implications for understanding of the potent immunosubversive action of CyaA on phagocytes and of its central role in *B. pertussis* virulence and escape from innate immunity control. Despite the fact that CyaA was repeatedly reported to subvert FcR- and CR3-

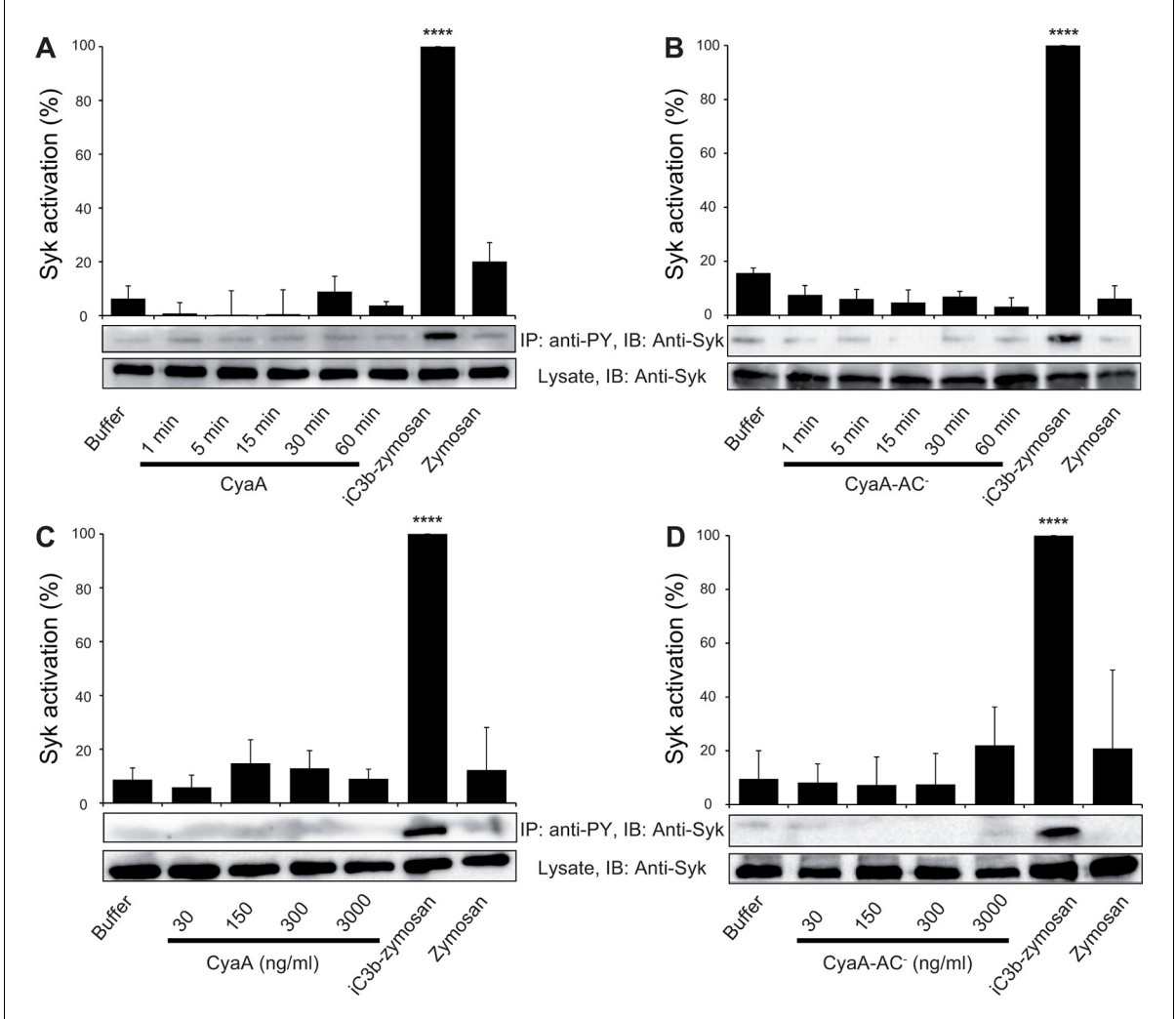

**Figure 6.** CyaA binding to CR3 does not trigger Syk activation. (**A, B**) $3\times10^6$ THP-1 cells were incubated with 30 ng/ml of CyaA (**A**), or CyaA-AC[-] (**B**) for indicated times. (**C, D**) $3\times10^6$ THP-1 cells were incubated with different indicated concentrations of CyaA for 15 min (**C**), or CyaA-AC[-] for 30 min (**D**). (**A-D**) Treated cells were lysed and cell lysates were immunoprecipitated (IP) with anti-phosphotyrosine (anti-PY) mAb. Syk-P immunoprecipitated from whole cell lysates was detected by immunoblotting (IB) with anti-Syk mAb and normalized to total Syk detected in whole cell lysates. Cells treated with iC3b-opsonized zymosan were taken as a positive control and the cells treated with buffer, or unopsonized zymosan were taken as negative controls. Each bar represents the mean value with SD of three independent experiments. In comparison to buffer-treated cells, a significant increase of Syk activation was observed only in cells treated with iC3b-opsonized zymosan (****, $p<0.0001$; ANOVA).

The following figure supplement is available for figure 6:

**Figure supplement 1.** CyaA binds efficiently and specifically the THP-1 cells.

mediated opsonophagocytosis (*Confer and Eaton, 1982*; *Kamanova et al., 2008*; *Lenz et al., 2000*), little attention has been paid to the effector molecules and pathways targeted by CyaA-generated cAMP signaling. The Syk tyrosine kinase pathway, pinpointed here as the target of CyaA action, plays a central role in both FcR- and CR3-mediated phagocytosis, as it is activated immediately upon binding of immunoglobulin or iC3b opsonins to the respective FcR and CR3 receptors (*Crowley et al., 1997*; *Kiefer et al., 1998*; *Shi et al., 2006*). We show that CyaA bypasses the Syk-activating interaction with CR3 through a subversive preferential recognition of its bent, resting conformation and by engaging the CR3 structures outside of the I-domain of CD11b. In contrast to iC3b opsonin, binding of CyaA, or of its non-enzymatic CyaA-AC[-] toxoid, did not trigger any CR3-mediated activation of Syk. Strikingly, CyaA-catalyzed elevation of cAMP in phagocytes effectively

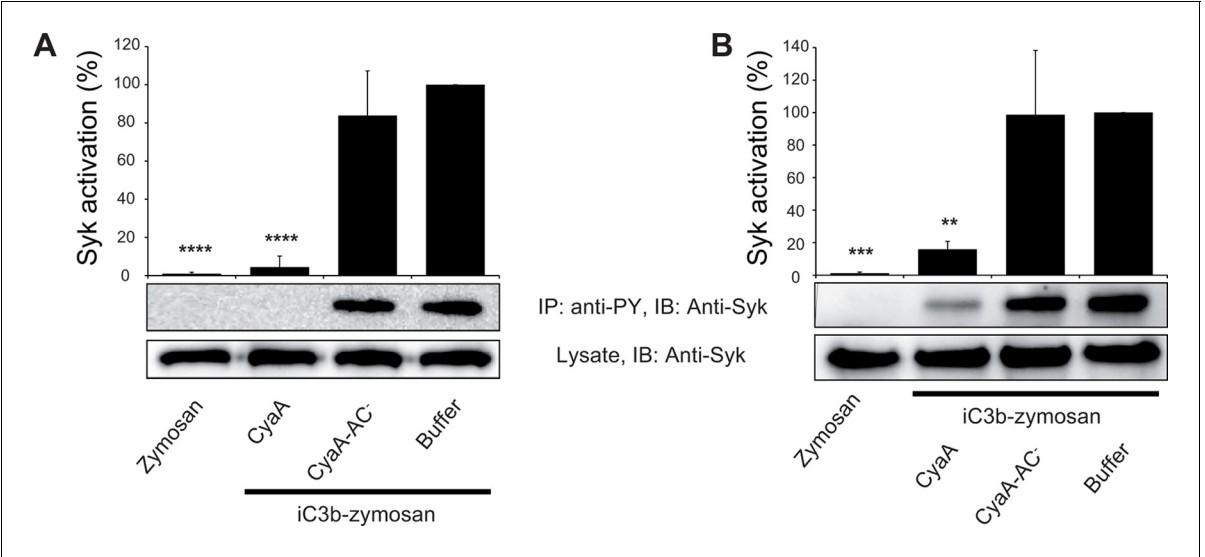

**Figure 7.** CyaA-produced cAMP blocks opsonin-induced Syk activation. (**A**) $3\times10^6$ THP-1 cells were incubated with 300 ng/ml of CyaA, CyaA-AC⁻ or buffer alone for 15 min and subsequently incubated with iC3b-opsonized zymosan for 30 min to activate Syk. Cells treated with buffer followed by unopsonized zymosan were used as a control. (**B**) $3\times10^6$ THP-1 cells were pre-incubated with iC3b-opsonized zymosan for 15 min to activate Syk and subsequently incubated with 300 ng/ml of CyaA, CyaA-AC⁻ or buffer alone for 30 min. Cells treated with unopsonized zymosan followed by buffer were taken as negative control. Processing of cells and detection of Syk were performed as in the legend to *Figure 6*. Each bar represents the mean value with SD of three independent experiments. Significant differences between mean values of Syk activation in cells treated with iC3b-opsonized zymosan in the absence or presence of CyaA or CyaA-AC⁻ are shown (\*\*, $p<0.01$; \*\*\*, $p<0.001$; \*\*\*\*, $p<0.0001$; ANOVA).

The following figure supplements are available for figure 7:

**Figure supplement 1.** CyaA and its enzymatically inactive variant CyaA-AC⁻ bind THP-1 cells with the same efficacy.

**Figure supplement 2.** CyaA and CyaA-AC⁻ do not alter the level of CR3 expression on the surface of THP-1 cells.

**Figure supplement 3.** CyaA reduces binding of iC3b-zymosan to THP-1 monocytes.

**Figure supplement 4.** Preincubation of THP-1 monocytes with iC3b-opsonized zymosan does not reduce binding of CyaA or of CyaA-AC⁻ to cells.

**Figure supplement 5.** CyaA and its enzymatically inactive variant CyaA-AC⁻ do not alter viability of THP-1 cells over the duration of the signaling experiments.

**Figure supplement 6.** CyaA-produced cAMP blocks opsonin-induced Syk activation.

blocked subsequent activation of Syk through iC3b binding to CR3. Most importantly, when Syk was pre-activated by incubation of phagocytes with iC3b, the subsequent CR3-dependent invasion of CyaA into cells enabled cAMP signaling-mediated inactivation of Syk (*Figure 8*).

It remains to be elucidated how CyaA-produced cAMP signaling suppresses Syk activity. Two plausible working hypotheses are, nevertheless, worth mentioning in this context. The first would be based on the observation that increased cAMP levels activate the inhibitory C-terminal Src kinase (Csk) via the cAMP-dependent protein kinase PKA (*Vang et al., 2001*). Csk may then directly inactivate the Src kinases by phosphorylation (*Vang et al., 2001*). Inactive Src kinases would then be unable to accomplish the double phosphorylation of the ITAM-containing adaptor, which is a prerequisite for recruitment and activation of Syk (*Jakus et al., 2007*; *Mócsai et al., 2006*; *2006*; *2010*; *Schymeinsky et al., 2007*). A second plausible mechanism would be based on our recent observation that CyaA/cAMP-mediated signaling triggers activation of the ubiquitously expressed SH2-containing non-receptor protein tyrosine phosphatase SHP-1 (PTPN6) (*Cerny et al., 2015*). This phosphatase has been previously implicated in regulation of tyrosine phosphorylation of

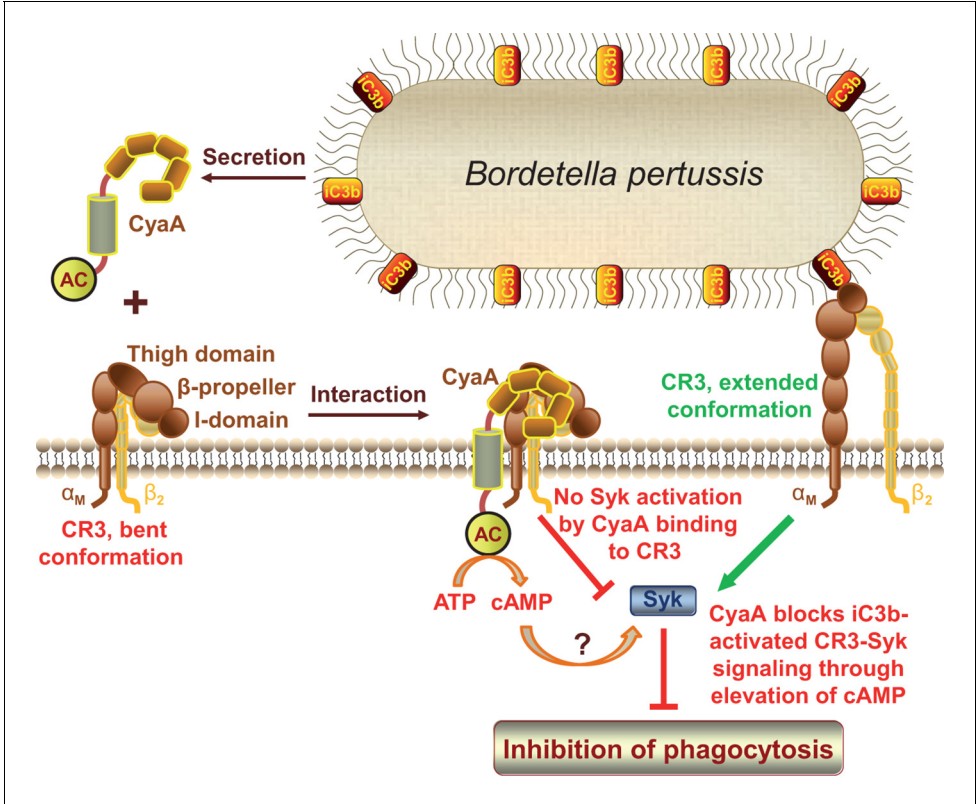

**Figure 8.** CyaA acts as a unique ligand of the I-domain-containing integrin CR3. CyaA secreted by *B. pertussis* binds CR3 outside of its I-domain, using a unique site that encompasses the C-terminal end of the last repeat of the β-propeller domain and the N-terminal portion of the thigh domain of CD11b. CyaA preferentially binds the integrin in a non-activated (bent, low-affinity) conformation and engagement of CR3 by CyaA does not trigger Syk activation in monocytes. Moreover, CyaA-catalyzed elevation of cAMP effectively blocks the iC3b opsonin-elicited activation of CR3-Syk signaling in monocytes. It remains to be elucidated how CyaA-produced cAMP signaling suppresses Syk activity. Binding outside of the I-domain in an activation-independent mode thus enables the toxin to hijack CR3 and block its signaling, thereby enabling *B. pertussis* to evade CR3-mediated phagocytosis.

Syk and in blocking of FcγR–mediated phagocytosis (*Huang, 2003*; *Kant et al., 2002*). Experiments aimed at deciphering the mechanism by which cAMP signaling inactivates Syk are ongoing.

CR3-mediated phagocytosis involves active RhoA GTP-ase, which acts downstream of Syk (*Caron and Hall, 1998*). We have previously shown that cAMP signaling of CyaA rapidly decreases RhoA activity and induces dephosphorylation of the actin filament-severing protein cofilin (*Kamanova et al., 2008*). CyaA thereby provokes massive actin cytoskeleton rearrangements and unproductive membrane ruffling of macrophage cells (*Kamanova et al., 2008*). We therefore speculate that this inhibitory intervention of the toxin at the two different signaling levels of Syk and RhoA, respectively, synergizes in bringing about a complete and rapid inhibition of CR3-mediated phagocytosis and complement-mediated opsonophagocytic killing of bacteria. CyaA would thereby subvert the central mechanism of innate defense against *B. pertussis* in naive hosts devoid of pathogen-specific antibodies, thus promoting bacterial colonization and virulence.

Besides FcR and CR3-mediated phagocytosis, Syk activity regulates a number of additional processes involved in the control of bacterial infections by the immune system. These include innate immunity mechanisms such as phagocyte chemotaxis, generation of reactive oxygen intermediates, neutrophil extracellular trap formation and degranulation of phagocytes (*Forsberg et al., 2001*; *Gevrey et al., 2005*; *Mócsai et al., 2002*; *Van Ziffle and Lowell, 2009*; *Willeke, 2003*). Moreover, Syk and its homologue, the ζ-chain-associated protein kinase of 70 kDa (ZAP-70), also play a crucial role in B- and T-cell receptor signaling during induction of adaptive immune responses (*Turner et al., 2000*). Indeed, chemotactic activity and oxidative burst of phagocytes, as well as the Th1/Th2 adaptive immune response balance, were both previously shown to be sensitive to the cAMP-elevating activity of CyaA (*Confer and Eaton, 1982*; *Friedman et al., 1987*; *Pearson et al.,*

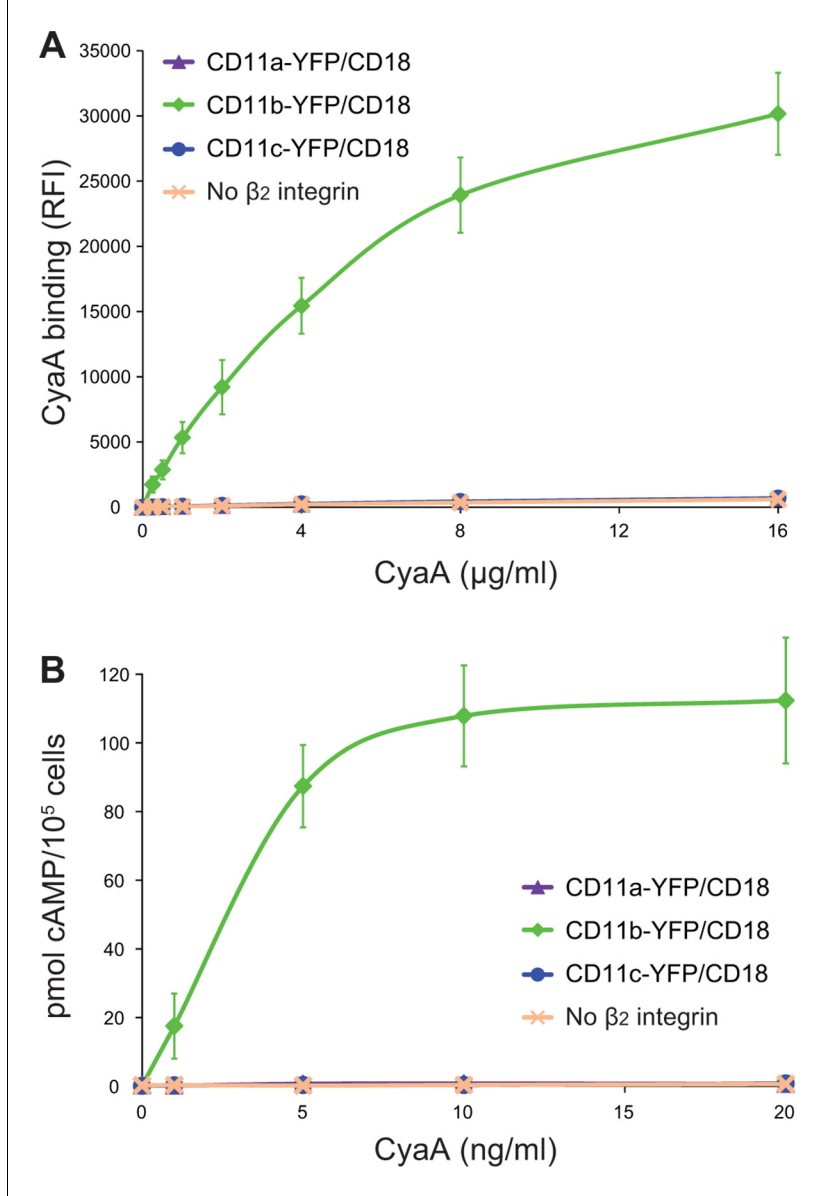

**Figure 9.** CyaA binds and intoxicates cells expressing LFA-1 with equally low efficacy as cells expressing CR4, or lacking any $\beta_2$ integrin at all. (**A**) $2 \times 10^5$ stably transfected CHO cells expressing CD11a-YFP/CD18, CD11b-YFP/CD18, CD11c-YFP/CD18, or no $\beta_2$ integrin were incubated with different concentrations of Dy647-labeled intact CyaA and analyzed by flow cytometry. Mean fluorescence intensities of CyaA binding were plotted against the concentrations of CyaA. RFI, relative fluorescence intensity. (**B**) $1 \times 10^5$ CHO cells expressing integrin molecules were incubated at indicated concentrations of CyaA and the amounts of accumulated cAMP were determined in cell lysates by ELISA. (**A** and **B**) Each point represents the mean value ± SD of three independent experiments performed in duplicate. CyaA binding to or cAMP intoxication of cells expressing CD11a-YFP/CD18, CD11c-YFP/CD18, or no $\beta_2$ integrin was at all toxin concentrations significantly lower than toxin binding to or cAMP intoxication of cells expressing intact CD11b-YFP/CD18 ($p < 0.0001$; ANOVA). However, CyaA binding to or cAMP intoxication of cells expressing CD11a-YFP/CD18 was at all measured toxin concentrations found to be statistically the same as toxin binding to or cAMP intoxication of cells expressing CD11c-YFP/CD18 or no $\beta_2$ integrin at all ($P > 0.1$; ANOVA).

The following figure supplements are available for figure 9:

**Figure supplement 1.** Expression of integrin variants fused with a fluorescent YFP protein on the surface of CHO cells.

**Figure supplement 2.** Residual binding of CyaA to CD11a-YFP/CD18 is not due to the presence of the YFP tag.

**Figure supplement 3.** CyaA does not interact with the intact native LFA-1 integrin.

*1987*; *Rossi Paccani et al., 2009*; *Weingart et al., 2000*). This suggests that disruption of Syk/ZAP70 signaling by CyaA impairs a broad range of processes that play a crucial role in host defense against infection.

Our demonstration of targeting and efficient blocking of Syk activation by CyaA-produced cAMP thus provides a novel mechanistic insight into the prominent role played by CyaA in mediating immune evasion of *Bordetella* species pathogenic to mammals. This underpins the crucial role played by the CyaA toxin in the early stages of host colonization by *B. pertussis* (*Goodwin and Weiss, 1990*; *Khelef et al., 1992*). It further indicates that vaccine-induced neutralizing antibody response against CyaA is likely to be importantly contributing to protection against infection and colonization by *B. pertussis*. Absence of the CyaA antigen from current acellular pertussis (aP) vaccines is thus of concern (*Sebo et al., 2014*). It may represent one of the factors contributing to recent resurgence of pertussis in the most developed countries that use the aP vaccine (*Cherry, 2012*; *Misegades et al., 2012*; *Octavia et al., 2012*; *Tartof et al., 2013*; *Witt et al., 2012*; *2013*).

## Materials and methods

### Antibodies

Monoclonal antibodies (mAbs) 2LPM19c (mouse IgG1) and 44 (mouse IgG1), specific for human CD11b, were obtained from Santa Cruz Biotechnology, Santa Cruz, CA. MAb M1/70 (rat IgG2b) specific for murine and human CD11b was purchased from BD Pharmingen. MAb ICRF44 (mouse IgG1) specific for human CD11b was obtained from BioLegend, San Diego, CA. MAb VIM12 (mouse IgG1) specific for human CD11b was obtained from Caltag Laboratories, Burlingame, CA. MAb 3.9 (mouse IgG1) specific for human CD11c was obtained from Ancell Corporation, Bayport, MN. MAbs specific for human CD11b (MEM-174, mouse IgG2a) and CD18 (MEM-48, mouse IgG1) were a kind gift of V. Horejsi (Czech Academy of Sciences, Prague, Czech Republic). MAb OKM1 (mouse IgG2b) specific for human CD11b was purified from the OKM1 hybridoma obtained from the European Collection of Cell Cultures, Porton Down, UK. Anti-human CD11a (MEM-25, mouse IgG1), anti-human CD18 (MEM-148, mouse IgG1), anti-phosphotyrosine (P-Tyr-01, mouse IgG1) and anti-Syk (SYK-01, mouse IgG1) mAbs were purchased from Exbio, Vestec, Czech Republic. Anti-human CD14 (TÜK4, mouse IgG2a) mAb was obtained from Dako, Glostrup, Denmark. Mouse mAb 3D1 recognizing CyaA was a kind gift of E. Hewlett (University of Virginia, Charlottesville, VA). Monoclonal antibodies were unlabeled and/or conjugated with Alexa Fluor 488 (AF488), fluorescein isothiocyanate (FITC), R-phycoerythrin (PE), allophycocyanin (APC) or biotin, respectively. Cy5-conjugated goat anti-mouse IgG F(ab')$_2$ fragment (GAM-Cy5) was obtained from Jackson ImmunoResearch Laboratories, West Grove, PA. Horseradish peroxidase labeled anti-mouse IgG antibody was purchased from GE Healthcare, Piscataway, NJ.

### Cells and growth conditions

CHO-K1 Chinese hamster ovary cells (ATCC CCL-61) and THP-1 human monocytic cells (ATCC TIB-202) were obtained from the American Type Culture Collection (ATCC, Manassas, VA) and tested for the absence of mycoplasma contamination by Hoechst stain. CHO cells were grown in F12K medium (GIBCO Invitrogen, Grand Island, NY) supplemented with 10% fetal calf serum (FCS) (GIBCO Invitrogen, Grand Island, NY) and antibiotic antimycotic solution (ATB, 0.1 mg/ml streptomycin, 1000 U/ml penicillin and 0.25 μg/ml amphotericin; Sigma-Aldrich, St. Louis, MO). THP-1 cells were cultured in RPMI 1640 (Sigma-Aldrich, St. Louis, MO) supplemented with 10% fetal calf serum and antibiotic antimycotic solution.

### Ethics statement

Commercial anonymous human blood and leukopacks were purchased from the Blood Bank of Thomayer Hospital, Prague, Czech Republic, and informed consent was therefore not applicable. Handling of cells of human origin was performed in compliance with the quality and safety requirements of the Act No. 296/2008 Coll. and the decree No. 422/2008 Coll., and the used protocols were in accordance with internal guidelines of the Institute of Microbiology of the CAS, v. v. i.

## Plasmid constructs

pT7CACT1 was used for co-expression of *cyaC* and *cyaA* genes in production of recombinant CyaC-activated CyaA in *Escherichia coli* under the control of the isopropyl-β-D-thiogalactopyranoside-inducible *lacZ* promoter (*Osicka et al., 2000*). Oligonucleotide-directed PCR mutagenesis was used to construct pT7CACT1-derived plasmids for expression of CyaA mutant variants (CyaA-D1193A +D1194A+E1195A and CyaA-E1232+D1234A, respectively) harboring negatively charged aspartate and glutamate residues of the CD11b binding site of the toxin substituted with alanine residues.

CyaA-AC⁻ toxoid, unable to convert ATP to cAMP, was expressed from pT7CACT1-derived plasmid generated by placing a synthetic BamHI linker (5′-GGATCC-3′), encoding a dipeptide GlyPhe, into the EcoRV site between codons 188 and 189 of *cyaA*, as described previously (*Osicka et al., 2000*). pACTΔ385-828 was used to produce a deletion mutant of CyaA, CyaAΔH, lacking the hydrophobic segment between residues 385-828 (*Šebo and Ladant, 1993*).

Human cDNAs encoding CD11b and CD18 were a kind gift of D. Golenbock, Boston University School of Medicine, Boston, MA (*Ingalls et al., 1998*). Human cDNAs encoding CD11a and CD11c were purchased from Geneservice, Cambridge, UK. The cDNAs for CD11a, CD11b and CD11c were cloned into the pcDNA3 expression vector (Invitrogen, Carlsbad, CA) and the cDNA for CD18 into the pcDNA3.1/Zeo (+) expression vector (Invitrogen, Carlsbad, CA) under the control of the human cytomegalovirus immediate-early promoter.

For production of chimeric CD11b-CD11c molecules, silent mutations not altering the sequence of encoded proteins were introduced by oligonucleotide-directed PCR mutagenesis to generate restriction sites in homologous regions of the cDNAs for CD11b and CD11c. These restriction sites were then used to construct the chimeric CD11b molecules having different segments replaced by the corresponding portions of CD11c (*Figure 1A*), or to obtain the CD11b molecule with deleted I-domain (*Figure 1A*), or to construct a CD11c molecule harboring the 614-682 segment replaced by the CD11b counterpart segment (*Figure 3A*). All cDNAs for expression of CD11-derived proteins were constructed in the pcDNA3 vector.

To express CD11-derived proteins C-terminally fused to yellow fluorescent protein (YFP), a DNA fragment encoding YFP, originating from the plasmid pEYFP-C3 (Clontech, Ozyme, Paris, France), was fused in frame to the 3′-end of cDNAs (cloned in pcDNA3) encoding CD11a, CD11b, CD11c and CD11c$_{614-682b}$, respectively, using standard PCR and molecular cloning techniques.

Oligonucleotide-directed PCR mutagenesis was used to construct pcDNA3-CD11b-derived plasmids for expression of the CD11b mutant variants CD11b$_{R619A+K621A}$, CD11b$_{E625A+N627A+R629A}$, CD11b$_{K644A+K646A}$, CD11b$_{E650A+R652A}$, CD11b$_{K659+T661A}$ and CD11b$_{R662A+R664A+R666A}$, respectively.

To express the secreted ectodomain of the integrin CR3 (sCR3), the segments encoding transmembrane helices and the C-terminal cytoplasmic tails of CD11b (codons 1106 to 1152) and of CD18 (codons 700 to 769) were replaced by short DNA segments encoding a C-terminal 6 x His tag.

## Production, purification and labeling of CyaA

CyaA and its mutant variants were produced in the presence of the activating protein CyaC, using the *E. coli* strain XL1-Blue (Stratagene, La Jolla, CA) and the proteins were purified by a combination of ion exchange chromatography on DEAE-Sepharose and hydrophobic chromatography on Phenyl-Sepharose (*Osicka et al., 2000*). Endotoxin was removed by repeated washes of the toxin-bound resin with 60% isopropanol (*Franken et al., 2000*). The preparations used in signaling experiments thus contained less than 0.1 EU/1 μg of CyaA as determined by Limulus Amebocyte Lysate assay (QCL-1000, Lonza, Walkersville, MD).

On-column labeling of CyaA and its mutant variants was performed after the DEAE-Sepharose purification step. Protein samples were diluted 4-times in ice-cold 50 mM Tris-HCl, pH 8.0 containing 1 M NaCl and loaded on Phenyl-Sepharose beads. To label CyaA with biotin, the column was extensively washed with PBS (12 mM Na$_2$HPO$_4$, 2 mM KH$_2$PO$_4$, pH 7.4, 3 mM KCl and 132 mM NaCl) and the beads were resuspended in PBS containing NHS-Sulfo-LC-Biotin (Pierce, Rockford, IL) in a concentration to reach a biotin:CyaA molar ratio of ~20:1. Biotin coupling was performed at 25°C and was stopped after 40 min by washing of the resin with 50 mM Tris-HCl, pH 8.0 solution, and then extensively with PBS. Purified biotinylated CyaA (CyaA-biotin) was then eluted with 50 mM HEPES, 8 M urea and 2 mM EDTA. To label CyaA and its mutant variants with Dyomics 647 (Dy647), the Phenyl-Sepharose column was washed with 50 mM sodium bicarbonate (pH 8.3) and the beads were

subsequently resuspended in the same buffer containing Dy647-NHS ester (Dyomics, Jena, Germany) in a concentration to reach a Dy647:protein molar ratio of ~6:1. Labeling was performed at 25°C for 2 hr, the column was washed with 50 mM Tris-HCl (pH 8.0) and labeled proteins were eluted in a buffer containing 50 mM Tris-HCl (pH 8) and 8 M urea. The binding, cell-invasive and hemolytic activities of labeled CyaA-derived proteins were determined as described elsewhere (*Osicka et al., 2000*) and were comparable (>90%) to activities of untreated CyaA.

### Establishment of stably transfected cell lines

A CHO cell line stably expressing the CD18 subunit (CHO-CD18) was established and used for cell surface expression of the CD11 subunits or their respective chimeric/mutant variants. Briefly, 0.8 µg of a highly purified plasmid DNA containing the appropriate integrin cDNA insert was mixed with 2 µl of Lipofectamine 2000 (Invitrogen, Grand Island, NY) in 100 µl of Opti-MEM I medium (Invitrogen, Grand Island, NY) and the mixture was incubated at 25°C for 20 min. The DNA:lipofectamine mix was added to CHO cells (> 90% confluency) in F12K medium with 10% FCS. After 6 hr at 37°C, the medium was changed, the transfected cells were cultured for 24 hr and then transferred to F12K medium supplemented with 10% FCS, ATB and 1000 µg/ml of G418 (InvivoGen, San Diego, CA) and/or 600 µg/ml of zeocin (InvivoGen, San Diego, CA) to select stable transfectants for 10 days.

Stably transfected CHO cells were stained with anti-CD11a, anti-CD11b, anti-CD11c and/or anti-CD18 mAbs labeled with AF488, FITC, or APC and cells expressing high levels of integrin molecules were selected, using a FACS Vantage cell sorter (Becton Dickinson, Franklin Lakes, NJ), by sorting single cells into individual wells of 96-well plates containing F12K medium supplemented with 10% FCS, ATB and 600 µg/ml of G418 and/or 300 µg/ml of zeocin. After 3 weeks, the cells from positive wells were expanded, stained with mAbs recognizing the integrin subunits and CHO clones expressing similar amounts of integrin molecules on the cell surface (the maximum difference between surface expression of mutant integrins and intact CR3 was ± 30%) were identified by flow cytometry on a FACS LSR II instrument (BD Biosciences, San Jose, CA). To monitor that the levels of integrin expression on the cell surface of CHO transfectants remained constant, the amounts of surface-expressed integrin molecules were systematically quantified in each experiment by staining with mAbs and subsequent flow cytometry analysis. The obtained mean fluorescence intensity (MFI) values from integrin detection were used to normalize the relative values of CyaA binding to cells.

To produce the secreted form of the integrin ectodomain (sCR3), CHO cells were co-transfected with plasmids harboring cDNAs encoding sCD11b and sCD18. A stably transfected cell line was established and individual cells were cloned from the bulk cell population using a FACS Vantage cell sorter. Stably transfected CHO-sCR3 cells were screened for secretion of the integrin ectodomain by a sandwich ELISA, using the MEM-48 mAb recognizing the CD18 subunit as a capture antibody and the biotinylated MEM-174 mAb directed against the CD11b subunit as detection antibody, respectively. The CHO-sCR3 clone with the highest secretion level of sCR3 was selected for further work.

### Production, purification and visualization of the CR3 ectodomain

CHO-sCR3 cells were cultured in F12 medium supplemented with 10% FCS at 37°C and culture supernatants were harvested every week. sCR3 was purified from the filtered culture supernatant by using an immunoaffinity chromatography matrix prepared by a covalent linkage of MEM-174 mAb to Cyanogen bromide (CNBr)-activated Sepharose 4B beads (GE Healthcare, Piscataway, NJ). The bent and extended conformers of the sCR3 ectodomain were separated by gel filtration chromatography on Superdex 200 HR (GE Healthcare, Piscataway, NJ) in HBSS buffer complemented with 2 mM $CaCl_2$ and 2 mM $MgCl_2$. The freshly separated conformers of sCR3 were directly applied on glow-discharged carbon copper grids (*Benada and Pokorny, 1990*), stained with 0.75% uranyl formate, and visualized in a Philips CM 100 transmission electron microscope (FEI, Eidhoven, Netherlands) equipped with a MegaViewII slow scan camera controlled by AnalySis 3.2 software (Olympus Soft Imaging Solutions, Münster, Germany).

### CyaA and mAbs binding to cells expressing integrin molecules

All assays were performed in HEPES-buffered salt solution (HBSS buffer; 10 mM HEPES, pH 7.4, 140 mM NaCl, 5 mM KCl) complemented with 2 mM $CaCl_2$, 2 mM $MgCl_2$, 1% (w/v) glucose and 1% (v/v) FCS (cHBSS) in 96-well culture plates (Nunc, Roskilde, Denmark). For staining of integrin molecules

on cell surface by mAbs, $2x10^5$ transfected CHO cells were incubated for 30 min at 4°C in 50 µl of cHBSS buffer containing mAbs diluted according to manufacturer´s instructions. For CyaA binding assay, $2x10^5$ transfected CHO cells were incubated in 100 µl of cHBSS buffer containing 2 µg/ml of CyaA-biotin (the concentration giving approximately a half-maximal binding of the toxin to CHO cells expressing intact CR3) for 30 min at 4°C. The surface-bound CyaA-biotin was stained with strep-tavidin-PE (diluted 1:400; eBioscience, San Diego, CA) for 30 min at 4°C. After washing, cells were resuspended in HBSS and analyzed by flow cytometry in the presence of 1 µg/ml of propidium iodide, or Hoechst 33258. Data were analyzed using the FlowJo software (Tree Star, Ashland, OR) and appropriate gatings were used to exclude cell aggregates and dead cells. CyaA binding was expressed as percentage of toxin binding to CHO cells expressing the native form of CR3: ((MFI of cells expressing the mutant integrin - MFI of mock transfected CHO cells) / (MFI of cells expressing intact CR3 - MFI of mock transfected CHO cells)) x 100.

## Blocking of CyaA binding by mAbs

For blocking of CyaA binding to CR3 by mAbs, $2x10^5$ CHO-CD11b/CD18 cells were preincubated for 15 min at 4°C in the presence of saturating concentrations of different mAbs in 45 µl of cHBSS buffer. CHO-CD11b/CD18 cells incubated with isotype control mAbs or with cHBSS buffer alone were used as controls. 5 µl of CyaA-biotin was then added to the cells in the continuous presence of the mAbs to a final concentration of 2 µg/ml and incubation was continued for 30 min at 4°C. CyaA binding was determined by flow cytometry as described above and expressed as percentage of toxin binding to CHO-CD11b/CD18 cells treated without mAb: ((MFI of mAb-treated cells incubated with CyaA - MFI of mAb-untreated cells incubated without CyaA) / (MFI of mAb-untreated cells incubated with CyaA - MFI of mAb-untreated cells incubated without CyaA)) x 100.

## cAMP assay

$1x10^5$ CHO transfected cells were incubated with various concentrations of CyaA ranging from 0 to 20 ng/ml for 30 min at 37°C in D-MEM and the reaction was stopped by addition of 0.2% Tween-20 in 50 mM HCl. Samples were boiled for 15 min at 100°C, neutralized by addition of 150 mM unbuffered imidazole and cAMP levels were determined as described previously (*Karimova et al., 1998*). Due to the extremely high specific catalytic activity of the AC domain, the concentration of CyaA in the cAMP assay was two to three orders of magnitude lower than the toxin amount used in the CyaA binding assay.

## Binding of CyaA to the bent and extended conformation of CR3

A leukocyte-enriched fraction was prepared from fresh human blood of healthy donors by hypotonic lysis of red blood cells. Briefly, 1 ml of whole blood was diluted to 25 ml of ice-cold 0.2% NaCl solution, incubated for 30 s, and isotonicity of the solution was restored by addition of 25 ml of ice-cold 1.6% NaCl. The suspension was centrifuged for 6 min at 250 g and the pelleted leukocytes were resuspended in 0.5 ml of the pre-warmed DMEM medium. The cells were treated without or with 100 nM phorbol 12-myristate 13-acetate (PMA) or 1 mM $Mn^{2+}$ ions for 15 min at 37°C to activate CR3. 100 µl aliquots containing $2x10^5$ cells were immediately stained with OKM-1 mAb recognizing both CR3 conformations, or with MEM-148 mAb recognizing only the extended integrin conformation, or with different concentrations of Dy647-labeled CyaA, in a combination with anti-CD14 mAb. After 2 min, cells were analyzed by flow cytometry in the presence of 1 µg/ml of Hoechst 33258. Data were analyzed using the FlowJo software (Tree Star, Ashland, OR) and live monocytes, gated based on light-scatter characteristics and CD14-positivity (monocytes did not change the level of surface expression of CR3 after incubation with PMA/$Mn^{2+}$ and/or CyaA), were used for assessment of CyaA binding.

## Isolation and purification of intact LFA-1

Human peripheral blood mononuclear cells were isolated from leukopacks of healthy donors using the Ficoll-Paque Plus density gradient centrifugation and were lysed with 50 mM Tris-HCl (pH 7.4), 150 mM NaCl, 1% Triton X 100, containing a protease inhibitor cocktail (Roche Diagnostics GmbH, Mannheim, Germany). LFA-1 was purified from cell lysates using the anti-CD11a mAb MEM-25 coupled to CNBr-activated Sepharose 4B beads (GE Healthcare, Piscataway, NJ).

## Surface plasmon resonance (SPR)

SPR measurements were performed at 25°C using a ProteOn XPR36 protein interaction array system (Bio-Rad Laboratories, Hercules, CA). The freshly separated bent and extended conformers of the sCR3 ectodomain and the freshly purified LFA-1 integrin were diluted to concentrations of 1–20 µg/ml in 10 mM acetate buffer (pH 5.0) and immobilized on a GLC chip using a ProteOn amine coupling kit (Bio-Rad Laboratories) at a flow rate of 30 µl/min. Non-reacted activated groups were blocked by the injection of 1 M ethanolamine (pH 8.5). The subsequent SPR measurements were carried out in 10 mM HEPES, pH 7.4, 150 mM NaCl, 2 mM $CaCl_2$ and 0.005% Tween 20 at a flow rate of 100 µl/min for both the association and the dissociation phase of the sensograms. To minimize non-specific binding and mass transfer effects, three coupling concentrations of each sCR3 conformer (leading to refractive index changes of 400, 800 and 1200 RU) were tested. Initial experiments at each of the sCR3 coating concentrations were conducted with CyaAΔH at concentrations ranging from 20 to 320 µg/ml. The initial estimates of $k_{off}$ values then showed a concentration independence at all coupling levels, indicating that artifacts from analyte rebinding and mass transfer effects could be ruled out. Consequently, a coupling level of 1200 RU and flow rate of 100 µl/ml was chosen for all remaining experiments.

The CyaAΔH protein was serially diluted in running buffer to concentrations of 20, 40, 80, 160 and 320 µg/ml and injected in parallel ('one-shot kinetics') over the immobilized sCR3 and LFA-1 surface. Response curves were evaluated using the ProteOn Manager software (Bio-Rad Laboratories). The sensograms were corrected for sensor background by interspot referencing (the sites within the 6×6 array, which are not exposed to ligand immobilization but are exposed to analyte flow), and double referenced by subtraction of analyte (channel 1–5) using a "blank" injection (channel 6). The data were analyzed by global fitting of the response curves using a 1:1 Langmuir binding kinetics and a bivalent analyte model to determine the kinetic association and dissociation rate constants. The Langmuir-type model assumes the interaction between ligand (L) and analyte (A) resulting in a direct formation of the final complex (LA): $L + A \xrightarrow{ka,kd} LA$, where $k_a$ and $k_d$ are the association and the dissociation rate constants, respectively. The bivalent analyte model assumes two-step association process: $L + A \xrightarrow{ka1,kd1} LA + L \xrightarrow{ka2,kd2} LLA$, where the first binding event is described by $k_{a1}$ and $k_{d1}$, while $k_{a2}$ and $k_{d2}$ describe the association and dissociation of the second binding event, respectively. Fitting of the binding curves revealed that the bivalent analyte model described the interaction of CyaAΔH with immobilized sCR3 significantly better in terms of reduced $\chi^2$ value and residual statistics than the simple 1:1 Langmuir binding model.

## Differential equations of the bivalent analyte model

$d[L]/dt = -(2 \times k_{a1} \times [L] \times [A] - k_{d1} \times [LA]) - (k_{a2} \times [LA] \times [L] - 2 \times k_{d2} \times [LLA])$

$d[LA]/dt = (2 \times k_{a1} \times [L] \times [A] - k_{d1} \times [LA]) - (k_{a2} \times [LA] \times [L] - 2 \times k_{d2} \times [LLA])$

$d[LLA]/dt = (k_{a2} \times [LA] \times [L] - 2 \times k_{d2} \times [LLA])$

| Parameter | Description | Fitting status |
|---|---|---|
| L | Concentration of free ligand (RU) | Constant |
| A | Concentration of free analyte (M) | Constant |
| LA | Concentration of ligand-analyte complex (RU) | Constant |
| LAA | Concentration of two ligand-analyte complex (RU) | Fit globally |
| $k_{a1}$ | Association rate constant for A + 2L ⇌ LA ($M^{-1}s^{-1}$) | Fit globally |
| $k_{d1}$ | Dissociation rate constant for LA ⇌ 2L + A ($s^{-1}$) | Fit globally |
| $k_{a2}$ | Association rate constant for LA + L ⇌ LLA ($RU^{-1}s^{-1}$) | Fit globally |
| $k_{d2}$ | Dissociation rate constant for LLA ⇌ LA + L ($s^{-1}$) | Fit globally |

## Fitting formulas of the bivalent analyte model

JacobN association

B = (Rmax-yJacob[0]-2*yJacob[1]);

dfdy[0][0] = -2*Ka1*Conc - Kd1 - Ka2*(B - yJacob[0]);

dfdy[0][1] = 2*(-2*Ka1*Conc + Ka2*yJacob[0] + Kd2);
dfdy[1][0] = Ka2*(B - yJacob[0]);
dfdy[1][1] = -2*(Ka2*yJacob[0] + Kd2);

## JacobN dissociation
B = (Rmax-yJacob[0]-2*yJacob[1]);
dfdy[0][0] = - Kd1 - Ka2*(B - yJacob[0]);
dfdy[0][1] = 2*(Ka2*yJacob[0] + Kd2);
dfdy[1][0] = Ka2*(B - yJacob[0]);
dfdy[1][1] = -2*(Ka2*yJacob[0] + Kd2);

## Derive association
B = (Rmax-yDerive[0]-2*yDerive[1]);
dydt[0] = B*(2*Ka1*Conc - Ka2*yDerive[0]) - Kd1*yDerive[0] + 2*Kd2*yDerive[1];
dydt[1] = Ka2*B*yDerive[0] - 2*Kd2*yDerive[1];

## Derive dissociation
B = (Rmax-yDerive[0]-2*yDerive[1]);
dydt[0] = B*(- Ka2*yDerive[0]) - Kd1*yDerive[0] + 2*Kd2*yDerive[1];
dydt[1] = Ka2*B*yDerive[0] - 2*Kd2*yDerive[1];

## Syk activation, immunoprecipitation and Western blotting

Experiments were performed on the human monocytic cell line, THP-1, and primary human monocytes isolated from leukopacks using the Ficoll-Paque Plus density gradient centrifugation followed by a magnetic cell-sorting using anti-CD14 magnetic beads to collect the CD14 positive fraction with an AutoMACS Pro Separator (Miltenyi Biotec, Bergisch Gladbach, Germany).

To opsonize zymosan with iC3b, the complement activation cascade in serum was used. Zymosan A (Sigma-Aldrich, St. Louis, MO) was incubated in 50% human serum at 37°C for 30 min. Zymosan particles were washed with PBS to remove unattached components of serum and dissolved in DMEM without FCS before use for treatment of cells. Unopsonized zymosan was prepared by incubating zymosan particles in PBS at 37°C for 30 min.

CyaA or genetically detoxified CyaA (CyaA-AC⁻) was diluted to final concentrations from concentrated stocks in DMEM without FCS. For the time course experiments, $3\times10^6$ THP-1 cells were treated with 30 ng/ml of toxin for different time points at 37°C. For concentration dependence experiments, $3\times10^6$ THP-1 cells were incubated at 37°C with different amounts of CyaA or CyaA-AC⁻ for 15 or 30 min, respectively. $3\times10^6$ THP-1 cells treated with 300 μg of opsonized or unopsonized zymosan were used as positive and negative controls, respectively. THP-1 cells treated with TUC buffer (50 mM Tris-HCl (pH 8.0), 8 M urea, 2 mM CaCl$_2$) diluted to the same level as the highest toxin concentration were used as an additional control.

To investigate the effect of elevated cAMP levels on Syk activation, $3\times10^6$ THP-1 cells or primary monocytes were pretreated with 300 ng/ml of CyaA, CyaA-AC⁻, or TUC buffer for 15 min, followed by treatment with iC3b-opsonized zymosan for 30 min at 37°C. Cells treated with TUC buffer followed by unopsonized zymosan were used as a control.

To examine the effect of cAMP on already activated Syk, $3\times10^6$ THP-1 cells or primary monocytes were incubated for 15 min with iC3b-opsonized zymosan followed by addition of 300 ng/ml of CyaA, CyaA-AC⁻, or TUC buffer for 30 min at 37°C. Cells incubated with unopsonized zymosan followed by TUC buffer were used as a control.

The reactions were carried out in DMEM with 10% FCS and no antibiotics. Triton X lysis buffer (50 mM Tris-HCl (pH 7.4), 150 mM NaCl, 1 mM Na$_3$VO$_4$, 10 mM NaF, 0.13% SDS, 1% Triton X 100 and EDTA-free protease inhibitor cocktail (Roche Diagnostics GmbH, Mannheim, Germany)) was used to stop the reaction and to lyse the cells at the respective time points. Cell lysates were collected and debris was removed by centrifugation. Aliquots of lysates were taken to detect the total amounts of Syk in individual samples. Tyrosine phosphorylated proteins were immunoprecipitated using anti-phosphotyrosine mAb (P-Tyr-01) for 3 hr at 4°C and bound to protein A-coupled sepharose beads (GE Healthcare, Uppsala, Sweden) overnight at 4°C. The beads were washed in Triton X lysis buffer

and boiled with Laemmli loading buffer. The proteins were separated by SDS-PAGE and transferred onto a nitrocellulose membrane (Pall Corporation, Pensacola, FL). The membrane was blocked with 5% BSA in TBS-T (25 mM Tris-HCl (pH 7.4), 300 mM NaCl, 2.6 mM KCl, 0.3% Tween 20) for 60 min at 25°C and incubated with anti-Syk mAb (SYK-01) for 1 h at 25°C. Upon washing with TBS-T, the membrane was incubated with horseradish peroxidase-conjugated anti-mouse IgG antibody for 1 hr and washed with TBS-T. The blots were developed using SuperSignal West Femto maximum sensitivity substrate (Thermo Scientific, Rockford, IL) and the chemiluminescent signal was recorded using an ImageQuant LAS 4000 Imager (GE Healthcare, Uppsala, Sweden).

### 3D structure modeling of a CyaA-CR3 interaction

The 3D structure model of the CR3 complex was generated with the Modeler suite of programs (*Eswar et al., 2003*) using the known 3D structure of the highly homologous CR4 complex (PDB ID 3K72, chains A B; omitting the highly flexible I-domain and the C-terminal domains that are not deposited in the PDB file) (*Xie et al., 2010*). The structure of CR3 was then equilibrated to allow partial spatial rearrangement of the protein. Short (2 ns) MD simulation was performed using OpenMM (*Eastman and Pande, 2010*) Zephyr (*Friedrichs et al., 2009*) (code freely available on https://simtk.org/home/zephyr) implementing GPU accelerated version of GROMACS suite of programs (*Van Der Spoel et al., 2005*). Implicit solvation (GBSA, $\varepsilon$=78.3, "'accurate water"' with collision interval of 10.99 fs) in combination with parm96 force-field was used (*Kollman, 1996*). The initial structure was next optimized and refined by simulation at 300 K with time step of 2 fs.

The structure of the amino acid segment 1166–1287 of CyaA, which was found to be responsible for CD11b binding (*El-Azami-El-Idrissi et al., 2003*), was predicted using I-TASSER (*Roy et al., 2010*).

To analyze the binding mode of CyaA to CR3, a flexible side chain docking of the modeled region 1166–1287 of CyaA to the CR3 complex was performed using the ClusPro server (*Comeau et al., 2007*).

The 3D structure of the full-length CD11b subunit was subsequently modeled using the structure prediction server I-TASSER (*Roy et al., 2010*). The resulting structure and orientation of the I-domain and of the C-terminal segment agreed with the crystal structure of CR4 (*Xie et al., 2010*).

### Statistical analysis

Results were expressed as the arithmetic mean ± standard deviation (SD) of the mean. Student's t-test was used to calculate statistical significance when two groups were compared. To test more than two groups, statistical analysis was performed by one-way ANOVA followed by Dunnett's post-test, comparing all the samples with the control. GraphPad Prism 6.0 (GraphPad Software) was used to perform statistical analysis. Significant differences are indicated by asterisks (*, $p<0.05$; **, $p<0.01$; ***, $p<0.001$; ****, $p<0.0001$).

## Acknowledgements

This work was supported by grants No. GAP302/11/0580 (RO), GA15-11851S (LB) and GA13-14547S (PS) from the Czech Science Foundation, by the Institutional Research Project RVO 61388971 of the Institute of Microbiology, by the project UNCE204025/2012 of the Charles University in Prague, by the project BIOCEV CZ.1.05/1.1.00/02.0109 (JC) from the ERDF and by the Institutional Research Project RVO 86652036 of the Institute of Biotechnology. The authors wish to gratefully acknowledge O Benada for expert technical help with the electron microscopy images and the excellent technical help of T Wald, H Kubinova and S Charvatova. SH is a doctoral student of the University of Chemistry and Technology, Prague.

## Additional information

### Funding

| Funder | Grant reference number | Author |
|---|---|---|
| Grantová Agentura České Republiky | GAP302/11/0580 | Radim Osicka |

| Institutional Research Project of the Institute of Microbiology | RVO 61388971 | Radim Osicka<br>Adriana Osickova<br>Shakir Hasan<br>Ladislav Bumba<br>Peter Sebo |
|---|---|---|
| Project of the Charles University in Prague | UNCE204025/2012 | Adriana Osickova |
| Grantová Agentura České Republiky | GA15-11851S | Ladislav Bumba |
| ERDF | BIOCEV CZ.1.05/1.1.00/02.0109 | Jiri Cerny |
| Institutional Research Project of the Institute of Biotechnology | RVO 86652036 | Jiri Cerny |
| Grantová Agentura České Republiky | GA13-14547S | Peter Sebo |

The funders had no role in study design, data collection and interpretation, or the decision to submit the work for publication.

### Author contributions

RO, Conception and design, Acquisition of data, Analysis and interpretation of data, Drafting or revising the article; AO, SH, LB, Conception and design, Acquisition of data, Analysis and interpretation of data; JC, Acquisition of data, Analysis and interpretation of data; PS, Conception and design, Drafting or revising the article

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
