## [Decision Letter]

Thank you for submitting your work entitled "*Bordetella* adenylate cyclase toxin is a unique ligand of the integrin complement receptor 3" for peer review at *eLife*. Your submission has been favorably evaluated by Richard Losick (Senior editor), a Reviewing editor, and three reviewers.

The reviewers have discussed the reviews with one another and the Reviewing editor has drafted this decision to help you prepare a revised submission. As you will see the major concerns are related to the methodology used to distinguish between active and inactive integrin conformations. The summaries of your work written by the three reviewers as well as the detailed concerns that need to be addressed are detailed below.

Essential revisions:

1) Figure 1 and experiments related to it.

The authors have overexpressed in CHO cells both α_M_ (CD11b) and α_X_ (CD11c) integrins and shown that CyaA only binds to α_M_. After that they have overexpressed numerous α_M_/α_X_ swaps.

The conclusion that CyaA cannot bind to α_M_ I-domain is based on the construct in which αI-domain region has been deleted.

The weakest CyaA binder was a receptor in which α_M_ contained α_X_ residues 614-682. These residues are said to locate "next to the β-propeller domain", but they are actually part of the "thigh" domain (In Figure 4 they have been correctly marked there). The structure and function of this domain should be discussed in more detailed manner.

The unconventional binding mechanism of CyaA is not a completely novel observation since Guermonprez et al. 2001 have reported that Mg^2+^ is not needed for CyaA recognition by α_M_β_2_ integrin. This fact should be discussed.

2) Figure 2.

The data based on antibodies support the existence of many binding site. Even the αI domain antibodies have a statistically significant effect. Can these antibodies activate/inactivate integrin conformation?

3) Figure 4.

The authors have used molecular modelling to indicate that it is structurally possible that CyaA binds to the thigh domain and propose that this takes place via electrostatic interactions. Based on the model they have done arginine to alanine mutations and indicate that simultaneous mutation of three arginine residues affects CyaA binding. Despite the fact that this observation suggests an interaction between CyaA and thigh domain (in addition to binding the β-propeller) more direct evidence is needed no really name the thigh domain as CyaA binding site. Could it be possible to produce α_M_ and α_X_ thigh domains as recombinant proteins?

4) Figure 5.

To test, whether CyaA binds to the bent or to the extended integrin conformation the authors have treated cells with an integrin activator PMA. PMA increases the number of activated integrins on cell surface and decreases CyaA binding. The observation is interesting, but one should remember that PMA has also many other effects on cells in general and, more specifically, on integrins. It may, for example, increase integrin clustering and mask binding sites in integrin legs. A better experiment would have been to use mutant integrins (e.g. E320A in α_M_) that favour the bent conformation together with integrins harbouring a gain-of-function mutation. Mn^2+^ could also have been used to activate integrins.

The authors have also isolated soluble α_M_β_2_ integrins and then managed to separate bent and extended conformations and to attach them to Biacore chips. It is quite surprising that the highly dynamic integrins can keep their conformations, but at least the data based on the conformation specific antibodies (Figure 5) supports this idea. However, the Biocore curves seem to be more or less identical (Figure 5) and indicate at best only marginal difference between the bent and the extended conformation. Table 1 shows the calculations based on bivalent analyte model. The authors should indicate a scientific reason why this model was chosen. They should also show the K_d_ values in addition to association and dissociation rates. How many times this experiment was repeated? Was the difference statistically significant, when independent experiments were combined?

The idea that the α_M_β_1_ integrin is in bent conformation is in conflict with a previously published report (Guermonprez et al. 2001), which indicates that Ca^2+^ is needed for the interaction and that EDTA and EGTA inhibit it. Depletion of Ca^2+^ forces the integrins to bent conformation (and should increase CyaA binding if it takes place to the non-activated integrin). The authors should be able to solve this critical discrepancy.

Interestingly, CyaA has been reported to bind α_L_β_2_ integrin that is in the activated conformation (Paccani et al. 2011). Is there a fundamental difference in the action of α_M_ when compared α_L_? Can this be explained by structural comparisons? Can α_L_ thigh domain bind to CyaA? Can α_M_/α_L_ I domains bind to CyaA?

The experiment in Figure 9 is suggesting that α_L_β_2_ could not recognize CyaA. Since this is in conflict with a previously paper, the authors should control the experiment very carefully and e.g. show that α_L_β_2_ is functional as YFP fusion protein. Does native α_L_ support CyaA entry in these assays?

5) Figure 6 and Figure 7 (Conclusion Figure 8).

The data nicely indicates that CyaA can inhibit iC3b induced phosphorylation of SYK. Based on the conclusion figure, one would also expect experiments testing whether CyaA prevents/allows α_M_β_2_ binding to iC3b? Similarly CyaA should not be able to affect recombinant α_M_ I-domain binding to iC3b.

6) Activation of the integrin with PMA decreases the capacity of CyaA to associate with these integrins. It would be interesting to know whether this is translated into a lower induction of cAMP by CyaA in target cells.

7) A weakness of that manuscript is the absence of numerical values for the respective K_d_ in the text. There are equations given in Materials and methods, a table, but these numerical values should be introduced in the text, for the deleted mutants, substituted mutants and various target forms developed in the manuscript.

Reviewer summaries:

Reviewer#1:

The adenylate cyclase toxin (CyaA) is a virulence factor of *Bordetella pertussis*. CyaA has been reported to bind to two integrin-type receptors on cell surface, namely α_M_β_2_ and α_L_β_2_, while a third integrin in the same subfamily, α_X_β_2_, cannot recognize CyaA. In this manuscript the authors show evidence that CyaA binds to the non-activated (bent) α_M_β_2_ state instead of the activated (extended) conformation. Moreover, they suggest that CyaA binding site is not in the αI (αA) domain. The exact significance of CyaA binding to cellular receptors is not clear (Its internalization is supposed to take place directly through plasma membrane, not via endocytosis, see Landant and Ullmann, 1999), but the observations are still interesting for integrin biology, since very little is known about the putative atypical ligand binding sites and mechanisms. Still, ligand binding to the bent integrin conformation is not entirely new idea, since non-activated integrins may participate e.g. in leukocyte rolling and human echovirus-1 binds to the non-activated α_2_β_1_ integrin.

The finding that CyaA binds to beta-propeller/thigh-domain in α_M_β_2_ and favour the bent integrin conformation would be important. However, based on the experiments shown in this manuscript, these conclusions are not fully justified. In some important points there are unsolved discrepancies when compared to previously published reports. In general the authors should study previous literature more carefully and in the discussion they should stress the structural aspects instead of signaling.

Reviewer#2:

The work presented by Osicka et al., describes the mode of interaction of the adenylate cyclase toxin CyaA from *Bordetella pertussis* with the integrin complement receptor 3 CR3 (CD11b/CD18) and how this modulates the activation of Syk-kinase downstream of the CR3.

By conducting two different approaches (CD11 b/c chimera and antibody interference) the authors first define that CyaA binds to the region 614-682 outside the I-domain. They also show that the swapping this stretch of amino acids between CD11b and CD11c confers to CD11c the capacity of binding to CyaA. The 3D structure modeling-coupled to mutagenesis followed by functional analysis allow the authors to identify in CyaA a group of negatively charged amino-acids implicated in this interaction and a group of positively charged amino-acids in their vicinity in CD11b. This convincingly points for the importance of electrostatic interactions in the interaction of the toxin with its receptor. They also established that CyaA binds preferentially to the close conformation / inactive form of the integrin. They also show that CyaA has a blocking effect on the activation of Syk prior to and after addition of iC3b-zymosan, in a AC independent and dependent manner, respectively.

This elegant work is well presented and the conclusions are thoroughly established by complementary approaches.

This work characterizes a new mode of interaction of a bacterial toxin with its integrin receptor that is critical to block Syk-dependent innate immune responses.

Reviewer #3:

Osicka and coworkers here reveal the major domains involved in the interactions between the Bordetella pertussis adenylate cyclase Cya and its major receptor CR3. Moreover, they dissected the signaling of a kinase Syk that is important in the CR3 signaling cascade to demonstrate that, after Cya integration into the cytoplasma, the elevation of cyclic AMP is responsible for the blockade of Syk phosphorylation. Overall, this is a huge work with many techniques, thoroughly conducted and abundantly explained and illustrated. There remains some weaknesses that should be easily correctly in order to render the text more substantial.

---

## [Author Response]

Essential revisions:

1) Figure 1 and experiments related to it.

*The authors have overexpressed in CHO cells both* α*_M_ (CD11b) and* α*_X_ (CD11c) integrins and shown that CyaA only binds to α_M_. After that they have overexpressed numerous α_M_/α_X_ swaps.*

The conclusion that CyaA cannot bind to α_M_ I-domain is based on the construct in which αI-domain region has been deleted. The weakest CyaA binder was a receptor in which α_M_ contained α_X_ residues 614-682. These residues are said to locate "next to the beta-propeller domain", but they are actually part of the "thigh" domain (In Figure 4 they have been correctly marked there). The structure and function of this domain should be discussed in more detailed manner.

We thank the reviewers for this point. The segment located between residues 614-682 encompasses, indeed, the C-terminal end of the last repeat of the β-propeller domain and the N-terminal portion of the thigh domain of CD11b. This is now appropriately described in the revised manuscript and the structural and functional features of the β-propeller/thigh region are described in more detailed manner in the Discussion section.

The unconventional binding mechanism of CyaA is not a completely novel observation since Guermonprez et al. 2001 have reported that Mg^2+^ is not needed for CyaA recognition by α_M_β_2_ integrin. This fact should be discussed.

As suggested, it is now mentioned in the revised manuscript that Guermonprez et al. 2001 observed Mg^2+^-independent CyaA binding to CR3 and suggested that CyaA might be an I-domain-independent ligand of the integrin.

2) Figure 2.

The data based on antibodies support the existence of many binding site. Even the αI domain antibodies have a statistically significant effect. Can these antibodies activate/inactivate integrin conformation?

To address the question of the reviewers, whether the I-domain-specific antibodies can activate/inactivate integrin conformation, the CHO cells expressing intact CR3 were pre-incubated with the 44 and 2LPM19c mAbs and the conformation of the integrin was probed with the MEM-148 mAb. As now shown in the newly added Figure 2—figure supplement 2, neither binding of these two mAbs, nor binding of any other mAb used in Figure 2 did cause any activation of the integrin. As clearly shown in Figure 1, the I-domain is not involved in the multivalent interaction of CyaA with CR3. Therefore, we conclude that the statistically significant decrease of CyaA binding to CR3, in the presence of the I-domain-specific mAbs 44 and 2LPM19c (Figure 2), resulted from steric hindrance of the 150 kDa-large I-domain-bound mAbs and not from activation of the integrin.

3) Figure 4.

The authors have used molecular modelling to indicate that it is structurally possible that CyaA binds to the thigh domain and propose that this takes place via electrostatic interactions. Based on the model they have done arginine to alanine mutations and indicate that simultaneous mutation of three arginine residues affects CyaA binding. Despite the fact that this observation suggest an interaction between CyaA and thigh domain (in addition to binding the β-propeller) more direct evidence is needed no really name the thigh domain as CyaA binding site. Could it be possible to produce α_M_ and α_X_ thigh domains as recombinant proteins?

We agree with the reviewers that demonstration of direct CyaA binding to the isolated segment 614-682 of CD11b (or the entire thigh domain) would potentially bring an added value. However, due to the complex and multivalent nature of CyaA interaction with CR3, such experiments are not likely to yield conclusive results. In contrast to binding of some endogenous ligands to the isolated recombinant I-domain of CR3 (Zhou et al., JBC, 1994), CyaA appears to recognize in a first step the presence of N-linked glycan chains of the C-terminal part of CD11b (Hasan et al. 2015), which is a critical prerequisite for toxin binding to CR3 (Morova et al. 2008). However, no N-glycan chains necessary for CyaA binding appear to be present within the segment 614-682 (Hasan et al. 2015). Moreover, as discussed in the manuscript and above, recognition of several segments of CD11b appears to contribute to the full capacity of CR3 to bind CyaA. Engagement of these segments likely facilitates or stabilizes the higher-affinity interaction of CyaA with the principal binding site located within the segment 614 to 682. Further, as now documented in the newly added Figure 1—figure supplement 3, CyaA recognizes the CD11b subunit only when this is in the heterodimeric complex with CD18. This strongly suggests that proper folding of the CD11b segment on the scaffold of the CD18 subunit is essential for toxin binding. This may explain why we failed to observe any significant inhibition of CyaA binding to the CD11b/CD18 heterodimer in the presence of a synthetic peptide derived from the 614-682 segment of CD11b (data not shown). Therefore, we opted here for a 'transplantation' experiment yielding positive evidence, where the 614-682 segment of CD11b was swapped into the context of the CD11c subunit of CR4 and conferred on CR4 a substantial capacity to bind CyaA and support toxin penetration into cells (Figure 3). We consider this as providing convincing evidence that the segment 614-682 is the principal binding site of CyaA. This conclusion is then strongly supported by loss of CyaA binding (i) upon swapping of the corresponding segment 614-682 of CD11c into CD11b (Figure 1); (ii) in the presence of mAbs competing with the toxin for the same segment (Figure 2), and (iii) upon site-directed mutagenesis of the segment 614-682 (Figure 4), respectively.

4) Figure 5.

To test, whether CyaA binds to the bent or to the extended integrin conformation the authors have treated cells with an integrin activator PMA. PMA increases the number of activated integrins on cell surface and decreases CyaA binding. The observation is interesting, but one should remember that PMA has also many other effects on cells in general and, more specifically, on integrins. It may, for example, increase integrin clustering and mask binding sites in integrin legs. A better experiment would have been to use mutant integrins (e.g. E320A in α_M_) that favour the bent conformation together with integrins harbouring a gain-of-function mutation. Mn^2+^ could also have been used to activate integrins.

As suggested by the reviewers, Mn^2+^ ions were used to activate CR3 on the surface of primary human monocytes. Similarly as with PMA, CyaA binding and cAMP intoxication of cells with the activated integrin was reduced in comparison with cells that were not treated with Mn^2+^ ions. The results addressing this issue are now shown in the newly added Figure 5—figure supplement 1. To control the possibility that activation of CR3 with PMA or Mn^2+^ ions would mask CyaA binding site in integrin legs due to increased integrin clustering, we used the OKM1 mAb that recognizes the same segment 614-682 of CD11b as CyaA. In contrast to reduced binding of CyaA to the activated CR3 integrin, no such reduction of binding was observed with the OKM1 mAb (Figure 5and Figure 5—figure supplement 1). The potential PMA-provoked integrin clustering thus does not appear to be involved in the masking of the CyaA binding site in integrin legs.

The authors have also isolated soluble α_M_β_2_ integrins and then managed to separate bent and extended conformations and to attach them to Biacore chips. It is quite surprising that the highly dynamic integrins can keep their conformations, but at least the data based on the conformation specific antibodies (Figure 5) supports this idea. However, the Biocore curves seem to be more or less identical (Figure 5) and indicate at best only marginal difference between the bent and the extended conformation. Table 1 shows the calculations based on bivalent analyte model. The authors should indicate a scientific reason why this model was chosen. They should also show the K_d_ values in addition to association and dissociation rates. How many times this experiment was repeated? Was the difference statistically significant, when independent experiments were combined?

The bent and extended conformers of secreted ectodomains of the β_2_ integrins CR4 and LFA-1 could be previously separated on the basis of their distinct hydrodynamic properties by gel permeation chromatography (Nishida et al., Immunity, 2006). Here we applied a similar procedure to isolate the bent and extended conformers of the soluble ectodomain of the β_2_ integrin CR3 (sCR3), documenting the separation by negative stain electron microscopy (Figure 5—figure supplement 2). It is important to note that the sCR3 separation, immobilization on an SPR chip and binding experiments were performed within a single day, in order to limit the extent of possible conformational changes of the purified sCR3 particles. The real time interaction of CyaA∆H with the bent and extended conformations of the sCR3 ectodomain revealed typical binding curves (Figure 5). These were fitted with several kinetic binding models (e.g. a simple 1:1 Langmuir model, a heterologous ligand model, a conformational change (two-state) model), where only the bivalent analyte model yielded satisfactory results in terms of reduced χ^2^ value and residual statistics that were not sensitive to changes in coupling levels and flow rates. As now stated in the legend to Table 1, two independent SPR binding experiments were performed in duplicate and the differences between mean values of CyaA∆H binding to the bent and extended conformation of sCR3 were statistically significant (P < 0.01; Student´s t-test). The determined K_d_ values of the interaction between CyaA∆H and the bent (K_d_ = 2.1x10^-7^ M) or extended (K_d_ = 6.4x10^-7^ M) conformers of sCR3 differed by a factor of ~3 and are now given in the revised text.

The idea that the α_M_β_1_ integrin is in bent conformation is in conflict with a previously published report (Guermonprez et al. 2001), which indicates that Ca^2+^ is needed for the interaction and that EDTA and EGTA inhibit it. Depletion of Ca^2+^ forces the integrins to bent conformation (and should increase CyaA binding if it takes place to the non-activated integrin). The authors should be able to solve this critical discrepancy.

The requirement for Ca^2+^ is not in conflict with preferential binding of CyaA to the bent conformer of the CR3 integrin. The reason is that CyaA is itself a calcium-binding protein and the functional folding of its receptor-binding RTX domain does strictly depend on loading of the ~40 Ca^2+^-binding sites formed by the RTX repeats of CyaA (Rose et al., JBC, 1995). Calcium loading then confers on CyaA a conformational change that enables it to bind and penetrate cells (Hewlett et al., JBC, 1991; Rogel et al., JBC, 1992). In the absence of Ca^2+^ ions, the CyaA cannot recognize the CR3 molecule because its RTX domain is not folded (Guermonprez et al. 2001).

*Interestingly, CyaA has been reported to bind α_L_β_2_ integrin that is in the activated conformation (Paccani et al. 2011). Is there a fundamental difference in the action of α_M_ when compared α_L_? Can this be explained by structural comparisons? Can α_L_ thigh domain bind to CyaA? Can α_M_/α_L_ I domains bind to CyaA? The experiment in Figure 9 is suggesting that α_L_β_2_ could not recognize CyaA. Since this is in conflict with a previously paper, the authors should control the experiment very carefully and e.g. show that α_L_β_2_ is functional as YFP fusion protein. Does native α_L_ support CyaA entry in these assays?*Already Guermonprez et al. 2001 have unambiguously demonstrated that CyaA binds selectively and with high affinity the cells expressing CR3, such as the dendritic cells and macrophages, whereas B and T cells, which express only the LFA-1 complex, are recognized by CyaA with very low efficacy, like other cells lacking CR3 and β_2_integrins, such as erythrocytes. Guermonprez et al. summarized their observations as follows: “The lack of efficient binding to CD11c/CD18 transfectants, or CD11a/CD18 expressing cells such as EL4 or LB27.4, also suggests that CD11b/CD18 is the only integrin of the β_2_ family involved in the binding of CyaA to the target cells”. We have fully reproduced here the findings of Guermonprez et al. 2001, showing that cells expressing LFA-1 bound about two orders of magnitude less CyaA toxin than cells expressing the same quantities of CR3, when the β_2_integrin expression level per cell was calibrated by using the YFP tag C-terminally fused to either of the alpha integrin subunits (Figure 9). In addition, there was no statistically significant difference in the levels of CyaA binding to cells expressing high amounts of CD11a-YFP/CD18, of CD11c-YFP/CD18, or not expressing any β_2_ integrin at all (Figure 9, CyaA binding to cells expressing CD11c-YFP/CD18 was newly added to the figure). This was not surprising, since CyaA has a well-documented capacity to bind and penetrate with a low efficacy the plasma membrane of all cell types lacking CR3 (reviewed in Vojtova et al. 2006; Ladant et al., Trends Microbiol, 1999; Hanski et al., Trends Biochem Sci, 1989).

To make sure that the low (residual) binding of CyaA to LFA-1 was not due to the presence of the YFP tag, we generated a cell line expressing comparable amounts of intact LFA-1 and demonstrated that CyaA binds cells expressing intact LFA-1 as poorly as cells expressing LFA-1 with the YFP tag (added as Figure 9—figure supplement 2). In addition, we purified the intact LFA-1 integrin from human peripheral blood mononuclear cells by immunoaffinity chromatography and immobilized it onto an SPR sensor chip. Real-time SPR measurements revealed that CyaA fails to bind the immobilized LFA-1 (added as Figure 9—figure supplement 3). Moreover, cells expressing LFA-1 were intoxicated by CyaA to equally low cAMP levels as the control cells expressing CR4, or lacking any β_2_ integrin. Indeed, cAMP intoxication levels of cells expressing CR3 were over two orders of magnitude higher than those of the control cells (added as Figure 9, panel B). This is in agreement with previously published data showing that a concentration of 5 ng/ml of CyaA was sufficient to produce 6 pmol of cAMP per 1 million of human THP-1 monocytes expressing CR3 (Ahmad et al., Cell Microbiol, 2015), while ~1600-times higher concentration of CyaA (8 μg/ml) had to be used to yield ~2 pmol of cAMP per 1 million of human blood T-cells or Jurkat T lymphoma cells expressing LFA-1 (Paccani et al. 2011). Moreover, LFA1-expressing Jurkat cells were previously found to bind CyaA as inefficiently as sheep erythrocytes that lack any β_2_ integrin (Gray et al., Infect Immun, 1999). Numerous in vitro studies previously reported that CyaA can penetrate a wide variety of cells lacking any β_2_ integrin, which is simply due to the extremely high specific AC enzyme activity. At >100 ng/ml concentration, the toxin causes easily detectable increase of intracellular cAMP concentrations over time in essentially any cell type (reviewed in Vojtova et al., Curr Opin Microbiol, 2006; Ladant et al., Trends Microbiol, 1999; Hanski et al., Trends Biochem Sci, 1989).

In conclusion, data obtained in several different laboratories show that CyaA binds and intoxicates cells expressing CR3 with substantially higher efficacy than cells expressing LFA-1. The data further demonstrate that CyaA does not bind LFA-1 any more avidly or in any more specific manner than it interacts through its weak lectin activity with other β_2_ integrin-unrelated glycoproteins or glycolipids of cell surface (Gordon et al., JBC,1989; Vojtova et al., MRT, 2006; Morova et al. 2008).

5) Figure 6 and Figure 7 (Conclusion Figure 8).

The data nicely indicates that CyaA can inhibit iC3b induced phosphorylation of SYK. Based on the conclusion figure, one would also expect experiments testing whether CyaA prevents/allows α_M_β_2_ binding to iC3b? Similarly CyaA should not be able to affect recombinant α_M_ I-domain binding to iC3b.

These experiments were, indeed, included as Figure 7—figure supplement 3 in the initial manuscript and demonstrated that pre-incubation of monocytes with CyaA causes a partial reduction of iC3b-coated particle binding to CR3-expressing cells. However, a significant decrease of iC3b binding was not observed upon preincubation of monocytes with the CyaA-AC^-^ toxoid that is unable to catalyze conversion of cytosolic ATP to cAMP. Moreover, as revealed by mAbs staining in Figure 7—figure supplement 2, pre-incubation with CyaA did not provoke any significant decrease of CR3 levels exposed on cell surface, showing that decreased binding of iC3b-coated particles was not due to CyaA-triggered CR3 internalization. In contrast, pre-incubation of cells with iC3b-opsonized zymosan did not reduce binding of CyaA or CyaA-AC^-^ (Figure 7—figure supplement 4). It indicates that the CyaA/CyaA-AC^-^ molecules do not directly compete with iC3b for binding to CR3. Instead, the cAMP signaling action of CyaA by an as yet unknown mechanism causes a partial decrease of the iC3b opsonin-binding capacity of CR3 on cell surface.

*6) Activation of the integrin with PMA decreases the capacity of CyaA to associate with these integrins. It would be interesting to know whether this is translated into a lower induction of cAMP by CyaA in target cells.*As suggested by the reviewers, intracellular levels of cAMP produced upon interaction of CyaA with PMA-activated primary monocytes were determined. The results demonstrate that intracellular cAMP levels produced by the toxin are significantly lower in PMA-treated cells than in untreated cells. The data were now added into Figure 5 as the panel C and the corresponding modifications have been made in the main text and in the legend to Figure 5, respectively.

7) A weakness of that manuscript is the absence of numerical values for the respective Kd in the text. There are equations given in Materials and methods, a table, but these numerical values should be introduced in the text, for the deleted mutants, substituted mutants and various target forms developed in the manuscript.

With the membrane-penetrating CyaA toxin used in this study, determination of the equilibrium constants (K_d_ values) for binding to the membrane-anchored CR3 receptor variants on cell surface cannot be performed in any meaningful way. This is because the binding of CyaA to the receptor is followed by irreversible insertion of the hydrophobic pore-forming domain and of the acyl chains of the toxin into cellular membrane, which occurs even at low temperature used in the binding assay (Rogel et al., JBC, 1992; El-Azami-El-Idrissiet al. 2003; Masin et al., Biochemistry, 2005). Membrane insertion withdraws the cell-bound toxin molecules from the association-dissociation equilibrium with CR3 and shifts the binding equilibrium to the right, yielding the very high apparent affinities (~10^-9^ M) reported by Guermonprez et al., JEM, 2001. Therefore, the interaction between CyaA and the various CR3 variants could not be expressed as numerical K_d_ values. Instead, the interaction of CyaA with cells expressing various CR3 mutants was quantified by flow cytometry and expressed as % of binding compared to cells expressing intact CR3.

Since the construction and selection of transfectants and subsequent production and affinity purification of the secreted integrin ectodomain is a rather lengthy and very laborious procedure, this was undertaken here only for the intact CR3. It would go far beyond the scope of the present work to produce and purify the ectodomains of all mutant CR3 variants developed in the manuscript. Therefore, only the K_d_ values of the interaction between CyaA and the bent and extended CR3 ectodomain conformers were determined by SPR. These values were now added into the revised manuscript.